# A projection specific logic to sampling visual inputs in mouse superior colliculus

**Katja Reinhard**[1,2,3†], **Chen Li**[1,2,3†], **Quan Do**[1,4], **Emily G Burke**[1,4], **Steven Heynderickx**[1,5], **Karl Farrow**[1,2,3,5*]

[1]Neuro-Electronics Research Flanders, Leuven, Belgium; [2]VIB, Leuven, Belgium; [3]Department of Biology, KU Leuven, Leuven, Belgium; [4]Northeastern University, Boston, United States; [5]IMEC, Leuven, Belgium

**Abstract** Using sensory information to trigger different behaviors relies on circuits that pass through brain regions. The rules by which parallel inputs are routed to downstream targets are poorly understood. The superior colliculus mediates a set of innate behaviors, receiving input from >30 retinal ganglion cell types and projecting to behaviorally important targets including the pulvinar and parabigeminal nucleus. Combining transsynaptic circuit tracing with in vivo and ex vivo electrophysiological recordings, we observed a projection-specific logic where each collicular output pathway sampled a distinct set of retinal inputs. Neurons projecting to the pulvinar or the parabigeminal nucleus showed strongly biased sampling from four cell types each, while six others innervated both pathways. The visual response properties of retinal ganglion cells correlated well with those of their disynaptic targets. These findings open the possibility that projection-specific sampling of retinal inputs forms a basis for the selective triggering of behaviors by the superior colliculus.

**\*For correspondence:**
karl.farrow@nerf.be

[†]These authors contributed equally to this work

**Competing interests:** The authors declare that no competing interests exist.

## Introduction

The nervous system is built from a large set of diverse neuronal cell types that work together to process information and generate behavior (*Zeng and Sanes, 2017*). Sets of connected neurons can be divided up into 'hard-wired' circuits that enable robust, stereotyped, reflex-like behavioral responses (*Chen et al., 2011*; *de Nó, 1933*; *Lundberg, 1979*), and flexible networks that modify their computations based on context and experience (*Dhawale et al., 2017*; *Rose et al., 2016*). Many innate behaviors rely on subcortical circuits involving the same sets of brain structures in different species (*Aponte et al., 2011*; *Gandhi and Katnani, 2011*; *Hong et al., 2014*; *Tinbergen, 1951*). In the visual system, it remains unclear to what extent these circuits have hard-wired rules linking their inputs with downstream targets (*Cruz-Martín et al., 2014*; *Ellis et al., 2016*; *Gale and Murphy, 2018*; *Gale and Murphy, 2014*; *Glickfeld et al., 2013*; *Liang et al., 2018*; *Morgan et al., 2016*; *Rompani et al., 2017*; *Roson et al., 2019*).

The output of the mammalian retina, the first stage of visual processing, consists of over 30 different ganglion cell types which can be distinguished by their dendritic anatomy, response properties, or molecular markers (*Baden et al., 2016*; *Bae et al., 2018*; *Dhande et al., 2015*; *Farrow and Masland, 2011*; *Levick, 1967*; *Martersteck et al., 2017*; *Roska and Werblin, 2001*; *Sanes and Masland, 2015*). Each ganglion cell type informs one or several brain areas about a certain feature of the visual world (*Ellis et al., 2016*; *Martersteck et al., 2017*). One of the major retinorecipient areas is the superior colliculus, which receives approximately 85% of the retinal outputs in rodents (*Ellis et al., 2016*; *Hofbauer and Dräger, 1985*; *Linden and Perry, 1983*; *Vaney et al., 1981*).

The rodent superior colliculus is a layered brain structure that receives inputs from all sensory modalities and targets various nuclei of the midbrain and brainstem. The superficial gray and the optic layer form the most dorsal layers of the superior colliculus and are primarily innervated by the

retina (*May, 2006*). These visual layers consist of several groups of neurons with diverse morphology, visual response properties and long-range targets that include the lateral pulvinar, lateral geniculate nucleus and parabigeminal nucleus. Each neuron of the superficial superior colliculus has been estimated to receive input from on average six retinal ganglion cells (*Chandrasekaran et al., 2007*). However, the different ganglion cell types that provide input to specific superior collicular output pathways have not been characterized. As a result, it is unknown whether the different output pathways of the superior colliculus have a common or different sets of retinal inputs, and consequently whether different visual inputs give rise to the different behaviors initiated by the colliculus (*Dean et al., 1989*; *Evans et al., 2018*; *Shang et al., 2018*; *Shang et al., 2015*; *Wei et al., 2015*; *Zhang et al., 2019*).

To determine the wiring rules underlying the integration of retinal information by different output pathways of the superior colliculus, we used a combination of transsynaptic viral tracing and molecular markers to specifically label the retinal ganglion cells at the beginning of two circuits: one targeting the parabigeminal nucleus (colliculo-parabigeminal circuit) and the second targeting the pulvinar (colliculo-pulvinar circuit). These two circuits were chosen as they are each directly involved in mediating orienting behaviors and are not major recipients of direct retinal input (*Shang et al., 2018*; *Shang et al., 2015*; *Wei et al., 2015*). Using quantitative analysis of the retinal ganglion cell morphology and comparison of the visual response properties in the retina and target nuclei, we found strong specificity in the routing of visual information through the superior colliculus.

## Results

### Transsynaptic tracing of retinal ganglion cells from targets of the superior colliculus

To determine if visual features are selectively sampled by two targeted output pathways of the mouse superior colliculus, we used rabies-based viral tools to label retinal ganglion cells innervating either the colliculo-parabigeminal or colliculo-pulvinar circuit. Three properties of the labeled ganglion cells were characterized. First, we reconstructed each cell's anatomy, with a particular focus on quantifying its dendritic depth profile within the retina. If available, this was combined with information about each cell's molecular identity based on labeling by different antibodies, and subsequently matched to cell types within the database of the Eyewire Museum (http://museum.eyewire.org; *Bae et al., 2018*). Finally, the visual response properties of a subset of labeled neurons were measured.

To perform these experiments, we injected the parabigeminal nucleus or lateral pulvinar (*Figure 1* and *Figure 1—figure supplements 1* and *2*) with herpes-simplex virus (HSV) expressing rabies-G, TVA and mCherry, and subsequently injected EnvA-coated rabies virus coding for GCaMP6s (EnvA-SADΔG-GCaMP6s) into the superficial layers of the superior colliculus (see Materials and methods). This transsynaptic viral infection strategy resulted in the expression of GCaMP6s in several dozen retinal ganglion cells per retina that specifically innervate the targeted circuit. To infect neurons projecting to the lateral pulvinar we utilized a floxed version of the HSV virus (hEF1a-LS1L-TVA950-T2A-RabiesG-IRES-mCherry) in combination with the *Ntsr1-GN209Cre* mouse line, which ensured labeling of wide-field neurons of the superior colliculus that project to the lateral pulvinar and not adjacent thalamic nuclei (*Gale and Murphy, 2018*; *Gale and Murphy, 2014*).

### Anatomy of retinal inputs to the colliculo-parabigeminal and colliculo-pulvinar circuits

The morphology of 658 ganglion cells innervating the colliculo-parabigeminal (n = 241) and colliculo-pulvinar (n = 417) circuit were extracted. The anatomy of labeled ganglion cells was recovered by staining the retinas with antibodies against GFP (binding to the GCaMP6s) and ChAT, an internal marker of depth formed by starburst amacrine cells (*Sanes and Masland, 2015*; *Sümbül et al., 2014a*). A semi-automated image processing routine was applied to high-resolution confocal image stacks of each ganglion cell that enables a precise quantification of their dendritic morphology (*Sümbül et al., 2014a*; *Sümbül et al., 2014b*). The cells showed a variety of morphologies:~7% had bistratified dendritic trees (n = 49), either co-stratifying with the ChAT-bands, or stratifying outside the ChAT bands (*Figure 1D*);~17% were mono-stratified with dendrites below the ChAT-bands

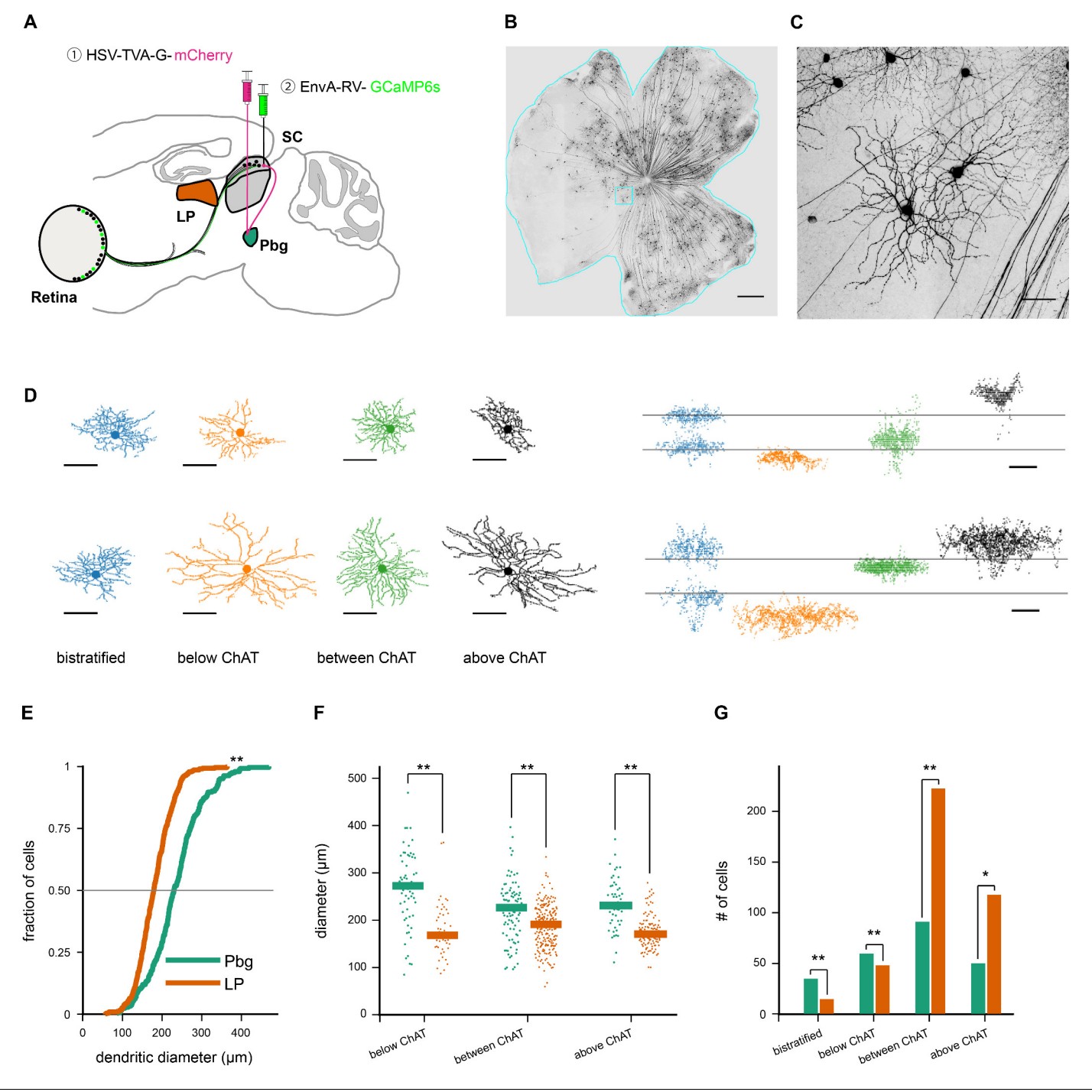

**Figure 1.** Transsynaptic tracing of retinal ganglion cells from the parabigeminal nucleus and the lateral pulvinar. (**A–C**). Labeling retinal inputs to the colliculo-parabigeminal circuit. (**A**) Injection strategy for labeling the circuit connecting the retina to the parabigeminal nucleus, via the superior colliculus. (**B**) Example retina with labeled ganglion cells innervating the colliculo-parabigeminal circuit. Scale bar: 500 µm. (**C**) Zoom into cyan box in B. Scale bar: 50 µm. (**D**) Eight example retinal ganglion cells from either injection approach (parabigeminal nucleus or pulvinar). Left: en-face view of the dendritic tree. Right: side-view of the dendritic tree. Location of the ChAT-bands is indicated with two gray lines. The cells are separated into four stratification groups: bistratified (first column), below (second column), between (third column), and above the ChAT-bands (last column). (**E**) Distribution of dendritic tree diameter of retinal ganglion cells that are part of the colliculo-pulvinar (LP; orange) and the colliculo-parabigeminal (Pbg; green) circuit. **p<0.01 Kolmogorov-Smirnov and Wilcoxon rank sum test. (**F**) Retinal ganglion cell diameters for cells stratifying below, between, and above ChAT-bands. **p<0.01 Kolmogorov-Smirnov and Wilcoxon rank sum test. (**G**) Retinal ganglion cells of each circuit were grouped into four

*Figure 1 continued on next page*

*Figure 1 continued*

stratification groups based on the peak of their dendritic profile. *p<0.05, **p<0.01 two proportion z-test. See also *Figure 1—figure supplements 1* and *2*.

The online version of this article includes the following figure supplement(s) for figure 1:

**Figure supplement 1.** Viral tracing with EnvA-coated rabies virus and herpes-simplex-virus (HSV).
**Figure supplement 2.** HSV injection sites of pulvinar and parabigeminal nucleus.

(n = 110);~50% had their dendrites restricted to the region between the ChAT-bands (n = 326); and ~26% had dendrites stratifying exclusively above the ChAT-bands (n = 173; *Figure 1*). We calculated for each cell the area covered by the dendrites and created a depth profile of the dendritic tree (*Figure 1—figure supplement 1*). Our data set contains cells with dendritic field diameters ranging from 57 to 468 μm (median: 194 μm), similar to the reported range of 80 to 530 μm (*Badea and Nathans, 2004*; *Bae et al., 2018*; *Coombs et al., 2006*; *Kong et al., 2005*; *Sun et al., 2002*).

Comparing the size and stratification of retinal ganglion cells innervating the colliculo-parabigeminal and colliculo-pulvinar circuits revealed two basic trends. First, cells innervating the colliculo-parabigeminal circuit had larger dendritic trees (median: 279 μm) than the cells innervating the colliculo-pulvinar circuit (median: 190 μm; p<0.01, Kolmogorov-Smirnov and Wilcoxon rank sum test; *Figure 1E*). This was true at each stratification level (*Figure 1F*). Second, the stratification depth of cells innervating each circuit had distinct distributions. While the colliculo-pulvinar circuit showed strong bias for sampling from neurons stratifying between (55.6%) and above (29.3%) the ChAT-bands, the colliculo-parabigeminal circuit sampled more evenly from each stratification level (bistratified 14.5%, below ChAT-bands 25.3%, between 39.0%, above 21.2%; *Figure 1G*). We found that these differences are not due to a bias in the retinotopic location of the sampled cells (*Figure 1—figure supplement 1*).

## Biased sampling of retinal ganglion cell types by the colliculo-parabigeminal and the colliculo-pulvinar circuit

To estimate the number of cell types innervating the colliculo-pulvinar and colliculo-parabigeminal circuits, we assigned our morphological data to one of the 47 putative retinal ganglion cell types documented in the Eyewire museum (http://museum.eyewire.org) (*Bae et al., 2018*). In addition we took into consideration information about genetically identified cell types including M2, sustained OFF-alpha cells, high-definition (HD)1, HD2, vertical OS cells and the four FOXP2[+] cells (*Jacoby and Schwartz, 2017*; *Nath and Schwartz, 2017*; *Nath and Schwartz, 2016*; *Rousso et al., 2016*; *Sümbül et al., 2014a*). The first step in our decision process was to find the most likely set of potential corresponding types based on stratification peak (above, below or between the ChAT bands) and, if available, molecular information (136/658 ganglion cells; n = 109 were SMI32[+]; n = 7 were CART[+]; n = 20 were FOXP2[+]). Second, the potential set of matching types was refined using a quantitative comparison of dendritic stratification profiles (see Materials and methods). Subsequently, we assigned each cell to its most likely cell type within the stratification based on quantifiable characteristics of the dendritic tree, as well as the shape and size of the soma (see Materials and methods for details). Finally, each cell was visually inspected to control for classification errors. Of the 47 cell types in the Eyewire museum, we were unable to reliably distinguish between a set of four pairs of cell types (1ni/1no, 4i/4on, 5ti/5I and 8n/9n), as well as the subtypes of direction-selective cells (ON-OFF: 37c,37d,37r,37v; ON: 7id,7ir,7iv,7o). This resulted in 37 possible cell types to which a ganglion cell could be assigned (see Materials and methods). Using this process, 599 of the 658 cells were assigned to one of 37 classes.

This analysis revealed that 14 of the 37 classes of retinal ganglion cells contained at least 1% of the ganglion cells from our data set, suggesting that a limited set of retinal ganglion cell types are sampled by the colliculo-pulvinar and colliculo-parabigeminal circuits (*Figure 2* and *Figure 2—source data 1*). These 14 putative cell types contain 550 out of the 599 classified cells and will subsequently be referred to as clusters 1–14, where the corresponding cluster in electron microscopy data of *Bae et al. (2018)* is referred to as EM C-xx (e.g. cluster 1 is EM C-1wt. *Figure 2*).

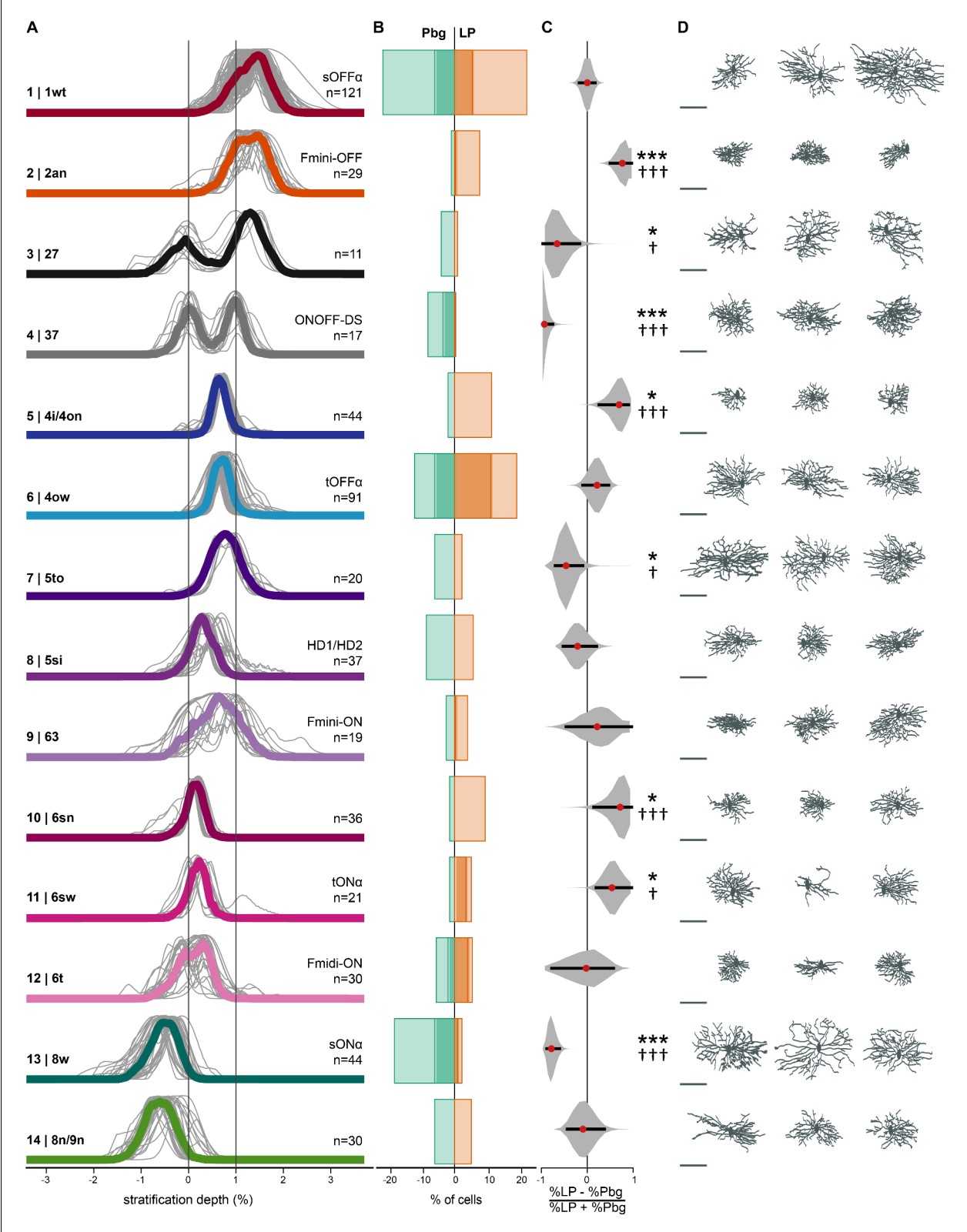

**Figure 2.** Retinal ganglion cell types targeting parabigeminal- and pulvinar-projecting collicular neurons. (A) Individual and median stratification profiles of 550 cells for the 14 cell types that innervate the colliculo-parabigeminal and colliculo-pulvinar circuit. Cluster names and additional names refer to the classification and assignment in *Bae et al. (2018)*. (B) Percentages of cells in each cluster for both circuits (100% equals the total number of cells in a given circuit, n = 196 for Pbg and n = 354 for LP). Darker shading represents the molecularly identified neurons. (C) Biases for the two circuits
*Figure 2 continued on next page*

*Figure 2 continued*

expressed as percentage differences. Black lines span 2.5% to 97.5% of the data. Red dots indicate the ratio calculated from B. *p<0.05, ***p<0.001, bootstrap analysis; † p<0.05, ††† p<0.01, two proportion z-test. (D) En-face view of example cells. Scale bar: 100 µm. See also *Figure 2—source data 1*.

The online version of this article includes the following source data for figure 2:

**Source data 1.** All 599 cells in their corresponding cluster.

For each cell type, we tested if there was a bias in the proportion of cells innervating one of the two circuits, quantified using a selectivity index (where 0 indicates equal sampling, and 1 or −1 indicate unique sampling by either the pulvinar or parabigeminal nucleus, respectively). A bootstrapping analysis was used to estimate confidence intervals and, together with a two-proportion z-test, determine if this selectivity measurement was different from 0 (see Materials and methods). Eight cell types showed a strong bias for innervating one of the colliculo-parabigeminal or colliculo-pulvinar circuits (*Figure 2*; bootstrap and two proportion z-test p<0.05). We found that clusters 3, 4, 7 and 13 preferentially innervated the colliculo-parabigeminal (cluster 3 p=0.0322 | 0.0221; cluster 4 p=0.0009 | $10^{-5}$; cluster 7 p=0.0483 | 0.0358; cluster 13 p=0.0009 | $10^{-9}$, bootstrap analysis | two-proportion z-test), and clusters 2, 5, 10 and 11 preferentially innervate the colliculo-pulvinar circuit (cluster 2 p=0.0009 | 0.0001; cluster 5 p=0.0217 | 0.0001; cluster 10 p=0.0483 | 0.0001; cluster 11 p=0.0241 | 0.0299, bootstrap analysis | two-proportion z-test). The remaining six cell types (cluster 1, 6, 8, 9, 12 and 14) were found to innervate both circuits more evenly (cluster 1 p=0.9933 | 0.9794; cluster 6 p=0.3717 | 0.0513; cluster 8 p=0.5350 | 0.2452; cluster 9 p=0.6640 | 0.4568; cluster 12 p=0.9933 | 0.9731; cluster 14 p=0.8462 | 0.7091, bootstrap analysis | two-proportion z-test).

The cell types preferentially innervating the colliculo-parabigeminal circuit include two bistratified cell types, clusters 3 and 4. Cluster 4 (EM C-37) consists of the $CART^+$, ON-OFF direction-selective cells. Little is known about cluster 3 (EM C-27), apart from their sluggish ON-responses to moving bar stimuli (*Bae et al., 2018*). In addition, the colliculo-parabigeminal circuit receives specific input from cluster 7 (EM C-5to), whose members have large dendritic trees (median diameter: 229 µm) that stratify between the ChAT-bands and exhibit weak ON and stronger OFF-responses to a moving bar (*Bae et al., 2018*). The fourth specific cell type is the $SMI32^+$, sustained ON-alpha cells of cluster 13 (EM C-8w).

The cell types preferentially innervating the colliculo-pulvinar circuit include clusters 2, 5, 10 and 11. Cluster 2 (EM C-2an) and 5 (EM C-4i/4on) consist of small OFF-cells (median diameter: 139 and 185 µm). The $FOXP2^+$ Fmini-OFF cells in cluster 2 (EM C-2an) stratify just above the ChAT-band, while the cells in cluster 5 (EM C-4i/4on) have their dendrites between the ChAT-bands. In addition, this circuit receives inputs from two ON-cell types (cluster 10 and 11). Both, the small cells of cluster 10 (EM C-6sn) and the $SMI32^+$, transient ON-alpha cells of cluster 11 (EM C-6sw) stratify above the ON-ChAT-band.

The two circuits share non-biased inputs from six cell types, clusters 1, 6, 8, 9, 12 and 14. These contain the two $SMI32^+$, OFF-alpha types, sustained OFF-alpha (cluster 1, EM C-1wt) and transient OFF-alpha (cluster 6, EM C-4ow). Further, they are innervated by the $FOXP2^+$ ON-cells, the Fmini-ON in cluster 9 (EM C-63) and the Fmidi-ON in cluster 12 (EM C-6t). The medium sized cells (median diameter: 191 µm) in cluster 8 (EM C-5si) stratify between the ChAT-bands and are potentially HD1 or HD2 cells (*Jacoby and Schwartz, 2017*). The final cluster 14 (EM C-8n/9n) consists of an ON-cell type that stratifies below the ChAT-bands.

## Retinal inputs to the parabigeminal and the pulvinar circuit differ in molecular signature

In our anatomical classification, we found different innervation patterns of alpha retinal ganglion cell types for the two circuits. To be able to trace the whole dendritic tree, cells were chosen for morphological analysis based on being separate from neighboring cells. To confirm the observed differences in circuit biases (*Figure 2*), we performed a survey of histological staining against molecular markers of ganglion cell types that was independent of how separated cells were. Here, we counted the number of double-positive cell bodies to establish the overall number of molecularly identified cells in each circuit and analyzed local confocal scans around the soma to determine the distribution

of alpha cell types. The four alpha cell types were labeled using the SMI32-antibody (*Bleckert et al., 2014*; *Coombs et al., 2006*; *Huberman et al., 2008*; *Krieger et al., 2017*; *Peichl et al., 1987*). We found that around half of all rabies-labeled cells innervating the two circuits are alpha-cells (colliculo-parabigeminal median: 42%, n = 3 retinas; colliculo-pulvinar median: 53%, n = 4 retinas; *Figure 3* and *Figure 3—figure supplement 1*). To identify which of the four alpha cell types innervate each circuit, we acquired local z-stacks of SMI32$^+$/GCaMP6s$^+$ double labeled neurons (n = 91 cells in three mice for the colliculo-parabigeminal circuit; n = 90 cells in three mice for the colliculo-pulvinar circuit). Each neuron was manually classified based on dendritic stratification depth: sustained ON-alpha cells have dendrites below the ChAT-bands; the transient ON- and transient OFF-alpha cells have dendrites between the ChAT-bands, and the sustained OFF-alpha cell has dendrites above the ChAT-bands (*Bleckert et al., 2014*; *Krieger et al., 2017*). Both circuits sample from sustained and transient OFF-alpha cells (parabigeminal vs pulvinar median: 13% vs 20% sustained; 32% vs 29% transient OFF-cells; 100% corresponds to all GFP$^+$ cells). In contrast, transient ON-cells mostly inner-vate the colliculo-pulvinar circuit (parabigeminal vs pulvinar median: 4% vs 17%; p<0.05 two propor-tion z-test), while sustained ON-cells are almost exclusively labeled in the parabigeminal circuits (parabigeminal vs pulvinar median: 10% vs <0.5%; p<0.05 two proportion z-test).

In our data set, the bistratified cells with dendritic density peaks aligned with the ChAT-bands strongly resemble the morphology of ON-OFF direction-selective cells (*Sanes and Masland, 2015*). In the mouse retina, there are four types of ON-OFF direction-selective ganglion cells, each respond-ing to one of the four cardinal directions. Three of the four types can be labeled with anti-CART anti-bodies (*Dhande et al., 2013*). We performed anti-CART histological staining in a subset of the retinas (*Figure 3*). Double-labeled neurons (GCaMP6s$^+$ and CART$^+$) are found almost exclusively after retrograde tracing from the parabigeminal nucleus (*Figure 3E*; median: 6.9% of all GCaMP6s-postive cells, range: 4.3% to 9.1%, n = 3 retinas). In the pulvinar experiments, a negligible percent-age of the labeled ganglion cells are CART$^+$ (*Figure 3E* and *Figure 3—figure supplement 1*; median: 1.3%, range: 0% to 2.1%, n = 6 retinas).

The percentages of CART$^+$ and SMI32$^+$ cells in each circuit from these experiments are consistent with the proportions observed in our single-cell analysis (*Figure 2*), where we found that 7% of all labeled cells in the colliculo-parabigeminal circuit were in cluster 4 (putative ON-OFF direction-selec-tive cells), and 44% were in clusters 1, 6, 11 and 13 (putative alpha ganglion cells). Similarly, in the colliculo-pulvinar circuit <0.1% of the ganglion cells were classified as ON-OFF direction-selective and 51% were alpha cells. Furthermore, the distribution of each alpha cell type between the two cir-cuits matches the distributions we found after anatomical classification, where the OFF types were innervating both circuits, while the transient ON-alpha cells showed a strong preference for the colli-culo-pulvinar and the sustained ON-alpha for the colliculo-parabigeminal circuit. We found that tran-sient OFF-alpha cells were underrepresented and sustained OFF-alpha cells overrepresented in our anatomical data set (*Figure 2*), compared to the molecular analysis (*Figure 3*). Despite the different proportions among the SMI32$^+$ cells, in both data sets, both OFF types are found to innervate each circuit.

## Functional properties of retinal ganglion cells support anatomical classification

To determine if the functional response properties of cells within a cluster are consistent, we per-formed two-photon targeted patch-clamp recordings from transsynaptically labeled neurons (*Fig-ure 4* and *Figure 4—figure supplement 1*). We presented each neuron with a set of visual stimuli that included the 'chirp' stimulus and a moving bar (*Baden et al., 2016*). For a subset of 48 of the recorded cells, we were able to retrieve the anatomy after recording. These cells are part of the data set that was assigned to anatomical cell types (*Figure 2*). In order to test if cells in a given ana-tomical cluster have consistent light responses, we used the 'chirp' stimulus which can be used to distinguish between different ganglion cell types (*Baden et al., 2016*; *Jouty et al., 2018*; *Roson et al., 2019*). For each of the clusters containing at least one of the 48 both traced and patched cells, we calculated the average response to the 'chirp' stimulus and used it as a template (see Materials and methods). We then assigned each of the remaining retrogradely labeled patched retinal ganglion cells without anatomical data (n = 75) to one of the anatomical clusters based on the similarity between the average response of the neuron and the templates, using three distance metrics (see Materials and methods). A cell was assigned to a cluster if at least two distance

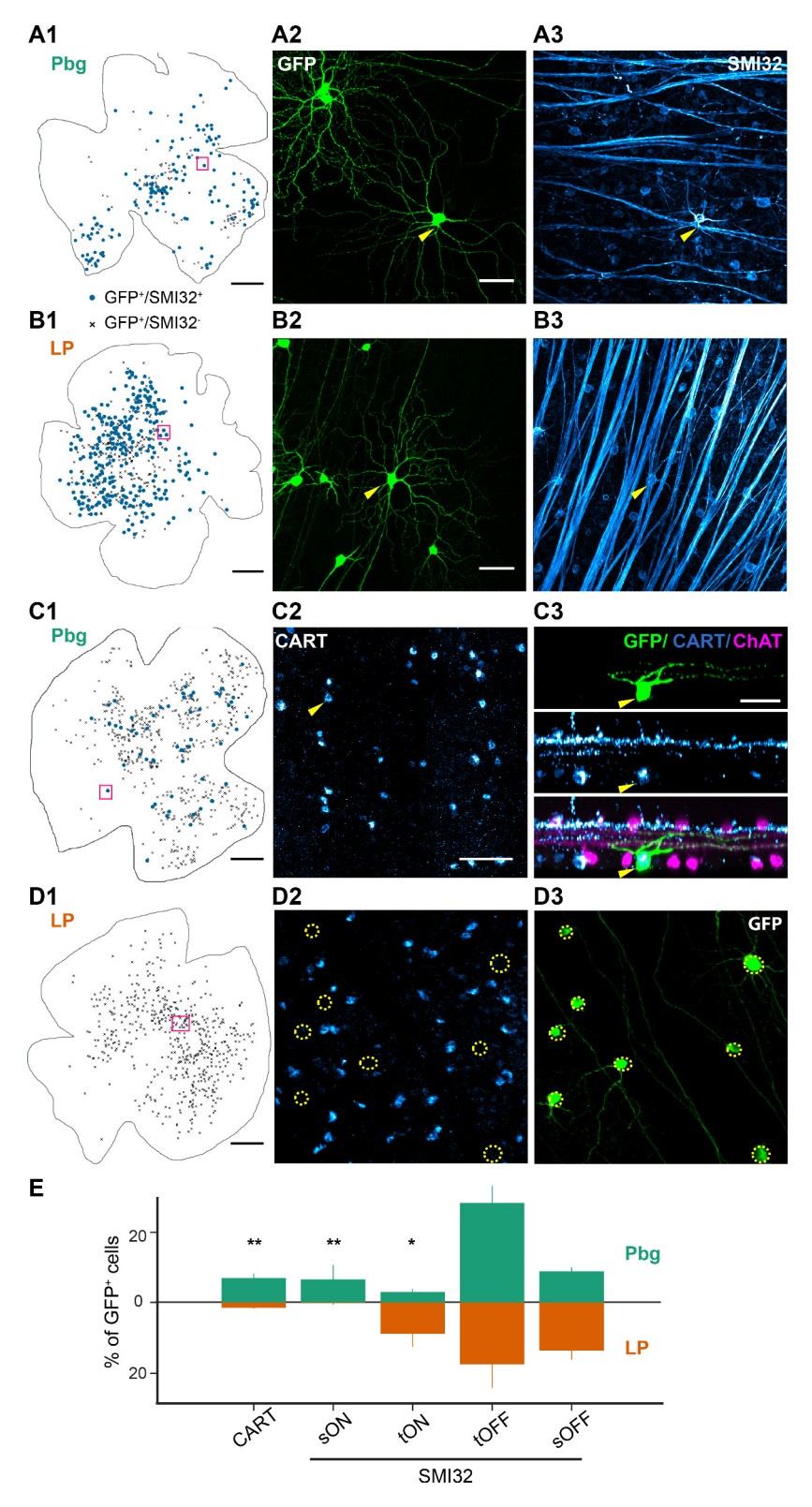

**Figure 3.** Distinct projection patterns of molecularly labeled retinal ganglion cells. (A–B) Example retinas with SMI32–positive labeled retinal ganglion cells innervating the colliculo-parabigeminal and colliculo-pulvinar circuit. (A1, B1) Example whole-mount view of the retina with SMI32-positive cells (blue dots) and SMI32-negative cells (other labeled retinal ganglion cells after virus injections; black crosses). Scale bar: 500 μm. (A2, B2) Histological

*Figure 3 continued on next page*

*Figure 3 continued*

staining against GCaMP6s. Yellow arrows indicate SMI32-positive retinal ganglion cells. (**A3, B3**) SMI32 histological staining against neurofilament. A2-3 and B2-3 are zoomed-in versions of the magenta square in A1 and B1. Scale bar: 50 µm. (**C–D**) ON-OFF direction-selective cells labeled with CART. Scale bar: 500 µm. (**C1, D1**) Example whole-mount view of the retinas with CART-positive (dots) and CART-negative (crosses) retinal ganglion cells. (**C2-3**) Histological staining against CART. Yellow arrows indicate a CART-positive retinal ganglion cell and its side-view. The cell has been labeled by the rabies virus (GFP-positive; top) and is CART-positive (middle). Bottom: overlay of GFP, CART, and ChAT-staining. C2-3 are zoomed-in versions of the magenta square in C1. (**D2-3**) No CART-positive neurons were labeled in the example retina from pulvinar experiments. D2-3 are zoomed-in versions of the magenta square in D1. Scale bar: 50 µm. (**E**) Percentage of CART-positive cells and the four different alpha ganglion cell types labeled in each circuit (100% corresponds to all GCaMP6s-expressing cells). Bars indicate standard errors. *p<0.05, **p<0.01 two proportion z-test. See also *Figure 3—figure supplement 1*. The online version of this article includes the following figure supplement(s) for figure 3:

**Figure supplement 1.** Number of GFP-positive neurons labeled by different molecular markers from colliculo-parabigeminal and colliculo-pulvinar circuit.

---

measurements ranked the cluster in the top two, if the distance measurements did not agree with each other, the cluster with the highest linear correlation coefficient was taken (see Materials and methods).

We report the visual responses for the seven cell types that contained at least four assigned cells (containing a total of n = 93 of the 123 patched cells with or without anatomy; n = 20 for cluster 1, n = 4 for cluster 4, n = 8 for cluster 5, n = 28 for cluster 6, n = 8 for cluster 7, n = 13 for cluster 11, n = 12 for cluster 13; *Figure 4*). We found that the visual responses to the 'chirp' stimulus were consistent with the predicted response based on published data (*Baden et al., 2016*) (*Figure 4B*). To test for responses to small moving objects we used a white bar moving with its short edge across the center of the cell's receptive field (*Figure 4C*). The average responses to the first, white and the second, black edge are consistent with published results for the different cell types (*Baden et al., 2016*; *Bae et al., 2018*). The single cell recordings to the 'chirp' and moving bar stimulus confirm that cells within a cluster show consistent visual responses, suggesting correct assignment of cells to the template clusters, and are consistent with the expected response properties of the anatomical cell type.

## Spatial distribution of cell types across the retina confirms correct classification

As an additional confirmation for the correct assignment of cells to the Eyewire clusters, we tested whether putative cell types showed the expected spatial distribution across the retina. For each of the retinal ganglion cell, we mapped its size onto its position within the retina. This revealed the expected general increase of dendritic size with distance from the optic nerve head for the whole ganglion cell population (*Figure 5A,E and I*). We then checked the spatial distribution of dendritic sizes of the sustained OFF and ON-alpha ganglion cells and the transient OFF-alpha cells (*Figure 5B–D*). It has previously been demonstrated that sustained ON and OFF-alpha cells, but not transient OFF-alpha cells show a strong asymmetric decrease of dendritic size along the naso-temporal axis (*Bleckert et al., 2014*). We found that the two sustained types show the previously reported negative correlation along the naso-temporal axis with small neurons found preferentially in the temporal retina (*Figure 5F–G*), and also replicated the equal distribution with respect to the distance from the optic nerve head (*Figure 5J–K*). As expected, we did not observe any asymmetry in dendritic size of transient OFF-alpha cells along the naso–temporal axis (*Figure 5H–L*) (*Bleckert et al., 2014*). In addition, while transient OFF-alpha cells do not show an asymmetric distribution of dendritic size, they do display an increasing response duration along the ventral – dorsal axis of the retina (*Warwick et al., 2018*). We found a similar increase in response duration for transient OFF-alpha neurons along this axis of the retina that was consistent, though with a weaker correlation, with *Warwick et al. (2018)* (*Figure 5M–P*). Taken together, these results demonstrate that our assignment of cell types based on anatomy are consistent with the known spatial distributions of anatomical and physiological characteristics of ganglion cells across the retina.

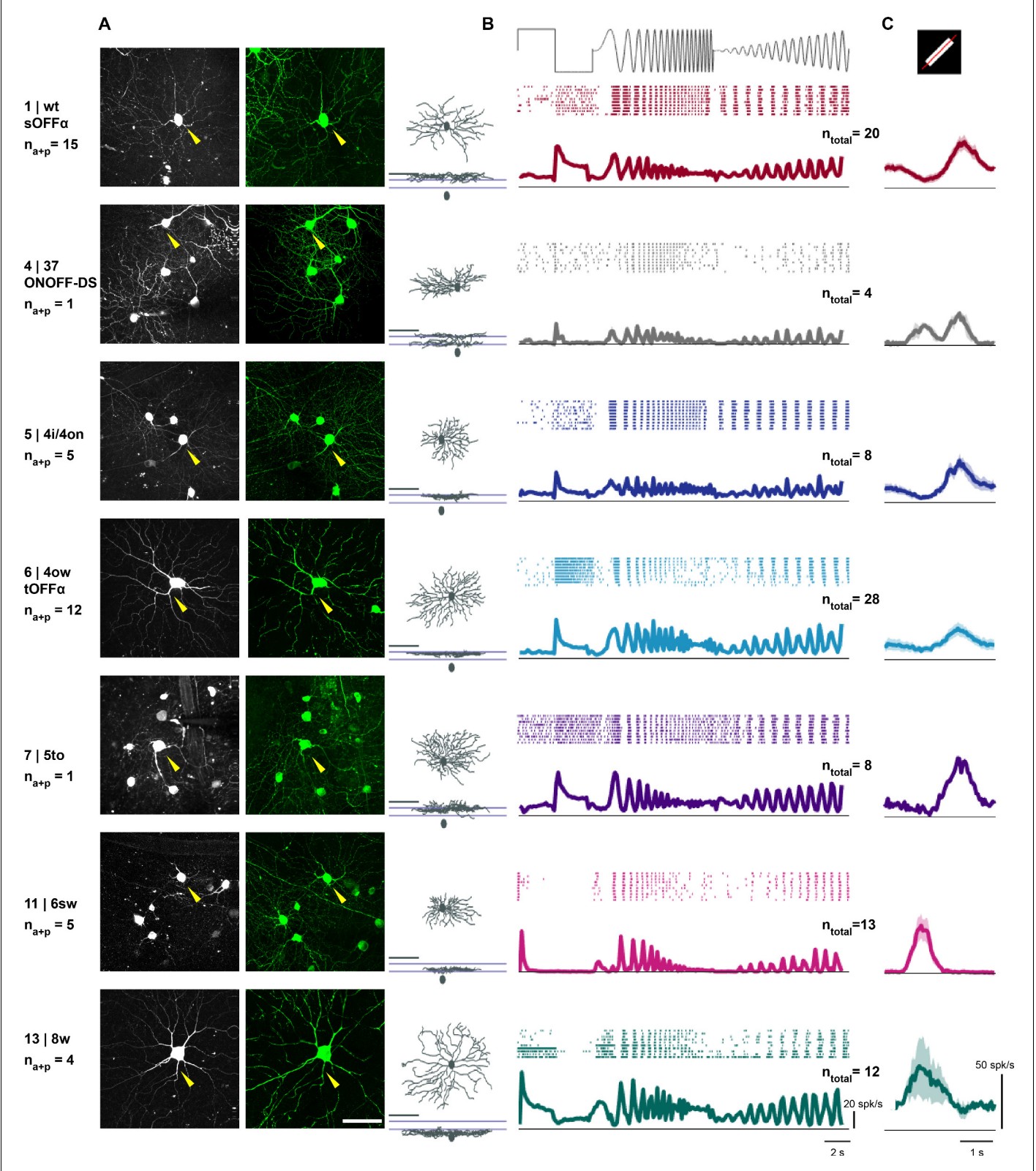

**Figure 4.** Visual response properties of anatomically identified retinal ganglion cells. **(A)** Left: maximum intensity projection of a two-photon image stack of a GCaMP6-expressing cell assigned to each cluster. Cluster number, assigned cluster in the Eyewire museum and the putative name are indicated in the left column. $n_{a+p}$ indicates the number of cells in this cluster with both anatomy and patch recordings. Middle: en-face view of a confocal microscope z-stack (maximum intensity projection) showing the same cell after staining (green: GCaMP6). Scale bar: 50 µm. Right: en-face and

*Figure 4 continued on next page*

*Figure 4 continued*
stratification view of this example cell after tracing. Scale bar: 100 µm. (**B**) Mean ± standard error of the mean (SEM) of responses to 'chirp' stimulus for cells assigned to an anatomical cluster. Top: raster plot of one example cell. $n_{total}$ indicates the number of cells with patch recordings. (**C**) Mean ± SEM of average response to a white bar moving in eight directions. The average response across all eight directions is plotted. See also ***Figure 4—figure supplement 1***.

The online version of this article includes the following figure supplement(s) for figure 4:

**Figure supplement 1.** Targeted patch-clamp recording of virus-labeled retinal ganglion cells.

## Some visual responses of pulvinar and parabigeminal nucleus are explained by selective innervation of retinal ganglion cell types

Taken together, the anatomical, physiological and molecular results indicate that different output pathways of the superior colliculus sample distinct sets of retinal inputs, where some inputs are biased towards a single pathway and others shared. We therefore asked if we could explain any of the response properties in the collicular targets by their preferential or shared sampling of ganglion cells.

To characterize the visual response properties of neurons in the pulvinar and parabigeminal nucleus, we performed single-unit recordings using Neuropixels high-density multichannel silicon probes (*Jun et al., 2017*) in awake, head-fixed mice (***Figure 6*** and ***Figure 6—figure supplement 1***). In each recording session, stereotaxic coordinates were used to target the parabigeminal nucleus or pulvinar. The recording locations were verified by histological reconstruction of the electrode tracts (***Figure 6B–C*** and ***Figure 6—figure supplement 1***). In the pulvinar, we only included recordings from its posterior portion, which receives input from the superior colliculus and does not respond well to full-field stimuli (***Figure 6*** and ***Figure 6—figure supplement 1***) (***Beltramo and Scanziani, 2019***; ***Bennett et al., 2019***). We recorded the brain activity on 384 electrodes spanning ~3800 µm in depth during visual stimulation (***Figure 6D and E***) and extracted the spikes from single units (***Figure 6F***). The receptive field centers of the recorded neurons were between −35° and +35° elevation and −65° and +25° azimuth.

We found that neurons in both the posterior pulvinar and parabigeminal nucleus responded reliably to a set of visual stimuli (***Figure 6G***) that includes: a large fast moving square ('big-fast', 53° side length, 150°/s); a small, fast moving spot ('small-fast', 4° diameter, 150°/s) and expanding discs (expanding from 2° to 50° of diameter within 300 ms). However, the percentages of responding units (maximal response > mean spontaneous firing rate + two std) differed for the different stimuli between the parabigeminal nucleus and the pulvinar (***Figure 6F*** and ***Figure 6—figure supplement 1***). Both responded to small, slow stimuli (diameter = 4°, speed = 21°/s) and expanding discs, however, more parabigeminal neurons responded to fast stimuli (150°/s) while the number of neurons responding to large, slow (size = 53°, speed = 21°/s) and dimming objects was larger in the pulvinar.

One key difference between the parabigeminal nucleus and pulvinar is their response to directional movement. We found strong and reliable responses of neurons in the parabigeminal nucleus to the presentation of a fast moving black square (53° side length, moving at 150°/s) moving in eight directions (***Figure 6G*** and ***Figure 6—figure supplement 1***). Only very few pulvinar neurons responded to this stimulus (***Figure 6G*** and ***Figure 6—figure supplement 1***); however, the response amplitude and duration of responding neurons were similar for both nuclei (***Figure 6—figure supplement 1***). A large fraction of parabigeminal neurons showed a preference for one or two directions of motion (***Figure 6I and K***). This direction-selectivity was not present in pulvinar neurons (***Figure 6J and K***; $p<0.05$ Kolmogorov-Smirnov test comparing the DSI distributions for the two nuclei). On the other hand, one key similarity between the two brain regions was their responses to a biologically relevant stimulus consisting of a black expanding disc (***Figure 6L–M*** and ***Figure 6—figure supplement 1***). Neurons in both the pulvinar and parabigeminal nucleus responded during the expansion phase of this stimulus, and there was no difference in the distribution of response strengths between the two populations of neurons ($p>0.05$ Kolmogorov-Smirnov test; ***Figure 6L–M***).

To gain insights into the response properties of the retinal ganglion cells innervating the colliculo-pulvinar and colliculo-parabigeminal circuits that might explain the similarities and differences recorded in the pulvinar and parabigeminal nucleus, we analyzed the visual responses of the 93

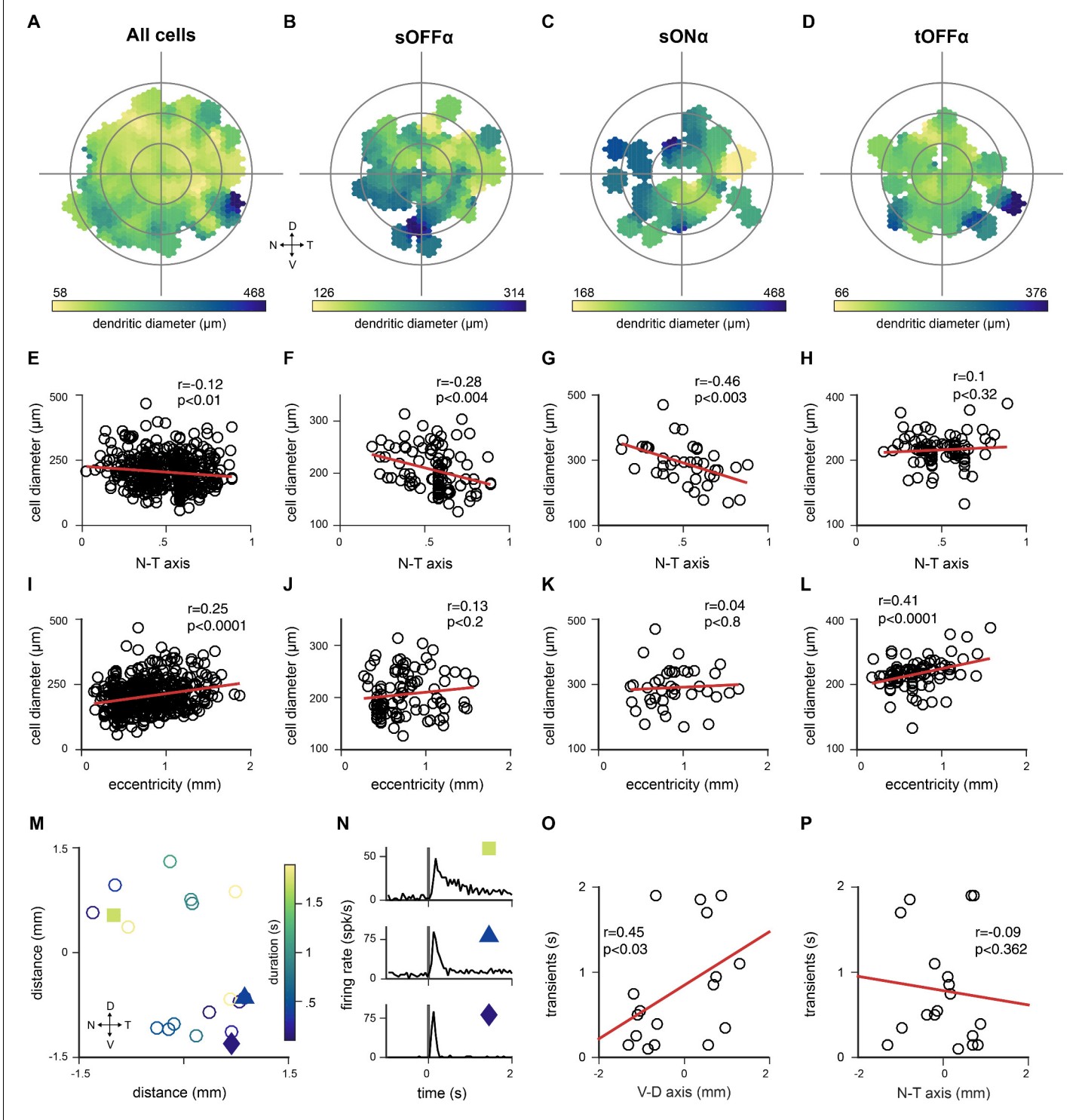

**Figure 5.** Spatial distribution of cell types across the retina confirms correct classification. (A–D) Distribution of dendritic field diameter of all labeled cells (A), sustained OFF (sOFF) alpha cells from cluster 1 (B), sustained (sON) alpha cells from cluster 13 (C) and transient OFF (tOFF) alpha cells from cluster 6 (D) at their retinotopic location. (E–H) Dendritic field diameter of all labeled cells (E), sOFF-alpha cells (F), sON alpha cells (G) and tOFF-alpha cells (H) are plotted along the nasal - temporal axis. sOFF- and sON-alpha cells showed negative correlation (sOFF: r = −0.28, p<0.004; sON: r = −0.46, p<0.003; two-tailed Pearson correlation coefficient test). (I–L) Dendritic field size of all labeled cells (I), sOFF-alpha cells (J), sON-alpha cells (K) and tOFF-alpha cells (L) relative to eccentricity (from optic nerve to periphery). tOFF-alpha cells and the whole labeled cell population showed positive correlation (tOFF: r = 0.41, p<0.0001; all cells: r = 0.25, p<0.0001; two-tailed Pearson correlation coefficient test) M) Positions of 18 labeled retinal ganglion cells that were assigned to the tOFF-alpha cell group. D, dorsal; V, ventral; T, temporal; N, nasal. Color bar indicated response duration time. *Figure 5 continued on next page*

*Figure 5 continued*

(N) Mean responses of 3 representative retinal ganglion cells from the tOFF-alpha cell group, whose locations are indicated in M. (O–P) Response durations are plotted across the ventral-dorsal axis (O) and nasal-temporal axis (P). Response transients gradually change along the ventral – dorsal axis (r = 0.45, p<0.03, Pearson correlation coefficient test).

retinal ganglion cells assigned to our anatomical clusters based on the response to the 'chirp' stimulus (see *Figure 4*). Fourier analysis of the responses to the frequency modulation part of the 'chirp' stimulus revealed a strong representation of low frequencies by cells in clusters 6 and 13 (*Figure 7A*), while cells in clusters 1, 5 and 7 showed a more even response profile for different frequencies. Cells in clusters 4 and 11 responded weakly to the full-field 'chirp' stimulus. We used the responses to a white bar moving in eight directions to evaluate the direction- and orientation-selectivity of each neuron. As expected, only the ON-OFF direction-selective cells in cluster 4 were direction-selective and none of the cell types showed evidence of orientation-selectivity (*Figure 7B*). Finally, we recorded responses to a black expanding disc (*Figure 7C*). The ON-cells in clusters 11 and 13 did not respond to this stimulus. The transient OFF-alpha cells (cluster 6), and cells in cluster 5 displayed the strongest responses during the period of expansion. The sustained OFF-alpha cells (cluster 1) and cells in cluster 7 responded later in the stimulus, with a peak firing rate as the disc reached its full size. The ON-OFF direction-selective cells (cluster 2) showed a biphasic response to the expansion stimulus.

Next, we combined our physiological data sets (*Figure 7*) with our assessment of the biases with which the different retinal ganglion cell types innervate the colliculo-pulvinar and colliculo-parabigeminal circuits (*Figure 2*) to ask if we could explain the similarities and differences we observed in the responses of neurons in the pulvinar and parabigeminal nucleus (*Figure 6*). This comparison revealed three clear relationships. First, both circuits receive input from cell types that respond during the expansion phase of the expanding disc stimulus (*Figure 2* and *Figure 7*). This included inputs from cluster 6 that innervates both circuits. These neurons are transient OFF-alpha cell, also known as the 'looming detector' (*Münch et al., 2009*). In addition, cluster 1, sustained OFF-alpha cells, innervate both circuits and respond during the presentation of expanding discs (*Figure 7*). This shared input to the colliculo-pulvinar and colliculo-parabigeminal circuits from neurons that respond to the presentation of dark expanding discs is matched by the shared response properties in the target nuclei (*Figure 6*). Second, in accordance with different direction-selectivity of the two target nuclei (*Figure 6*), we found that direction-selective retinal ganglion cells have a strong preference for the colliculo-parabigeminal circuit (*Figure 2*). Finally, a striking difference was observed between the responses of retinal ganglion cells innervating the different circuits and the responses of neurons in the target nuclei to full-field stimuli. The 'chirp' stimulus produces robust responses in most retinal ganglion cells but fails to illicit responses in either the posterior pulvinar, or the parabigeminal nucleus (*Figure 6—figure supplement 1*), which might be due to non-linear integration of retinal inputs or summation of opposite signed weights.

## Discussion

Comparing the morphological, molecular and visual response properties of retinal ganglion cells innervating the colliculo-parabigeminal and colliculo-pulvinar pathways passing through the superior colliculus has led to three conclusions (*Figure 8*). First, the colliculo-parabigeminal and colliculo-pulvinar circuit together sample from a limited set (14 of 37) of retinal ganglion cell types (*Bae et al., 2018*). Second, there is a clear preference in the set of retinal ganglion cell types providing input to each circuit. While four putative ganglion cell types show a strong preference for the colliculo-parabigeminal circuit, and four others for the colliculo-pulvinar circuit, six other types are more equally sampled by both circuits. Third, some response properties of neurons in downstream targets can be explained by the different and shared sampling biases of each retinal ganglion cell type by each collicular output pathway, respectively. These results support the notion that, in the superior colliculus, neural circuits are based on a dedicated set of connections between specific retinal inputs and different collicular output pathways.

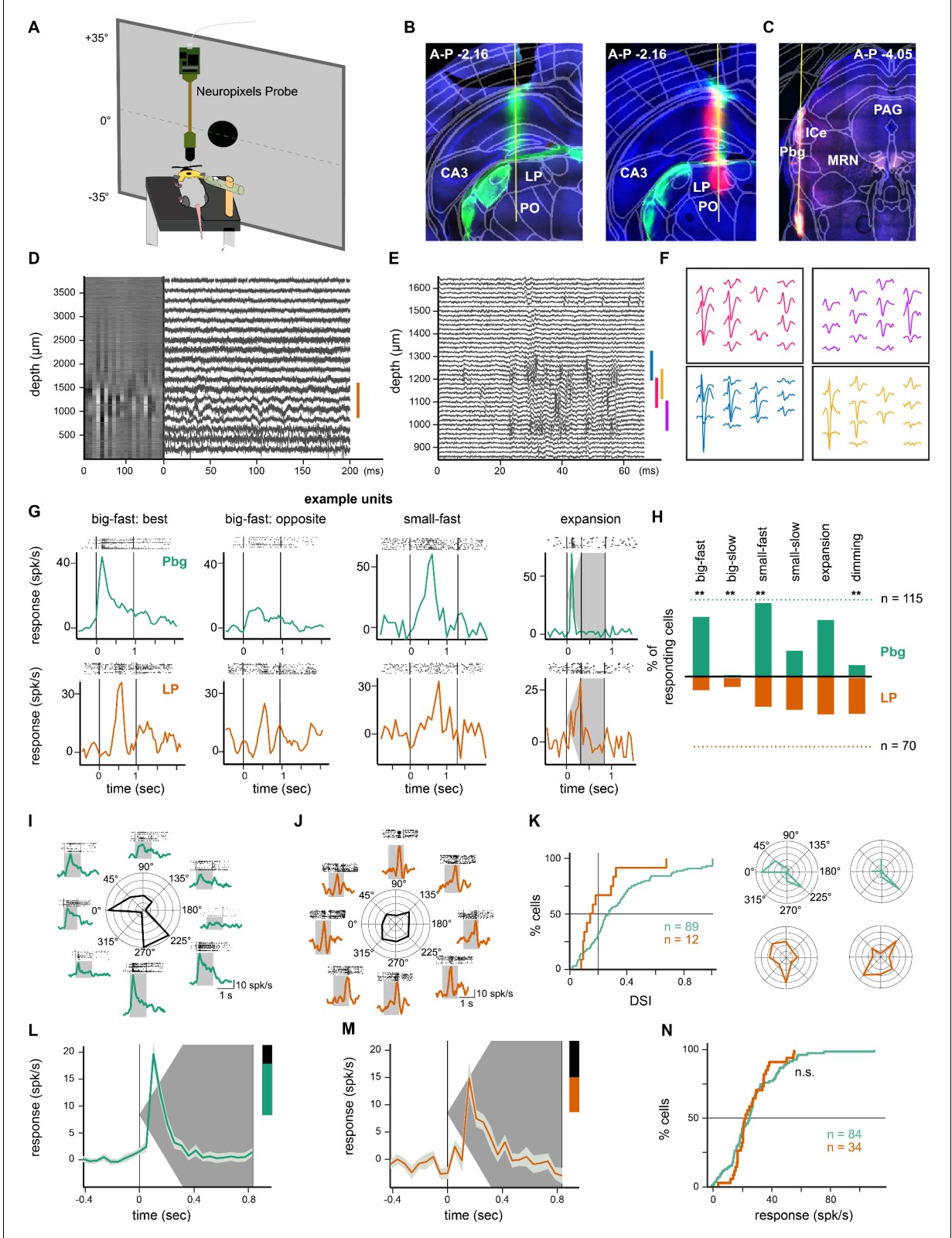

**Figure 6.** In-vivo recordings from the parabigeminal nucleus and pulvinar. (**A**) Schematic of the setup for Neuropixels recordings in awake, head-fixed mice. (**B**) Tracks of DiI- and DiD-coated probes (green and magenta) visible in the pulvinar. Retina targets, including the LGN, were labeled using Choleratoxin-b-Alexa488 injections into the eye (green). (**C**) Track of DiD-coated probe in parabigeminal nucleus. (**D**) Heatmap of activity on all 384 electrodes (300 Hz high-pass filtered data, 20 ms bins) and traces of example electrodes at different locations. The orange bar indicates the location of

*Figure 6 continued on next page*

Figure 6 continued

the pulvinar. (E) High-pass filtered activity only on the electrodes in the pulvinar during the presentation of an expanding disc. (F) Waveform footprints of four sorted neurons. Their location is indicated with colored bars in E. (G) Example responses from parabigeminal and pulvinar recordings to 10 repetitions of different stimuli. Stimuli were: Big-fast black square (53° side length, moving at 150°/s); small-fast black dot (4° diameter, moving at 150°/s); expanding black disc (expanded from 2° to 50° of diameter within 300 ms). The vertical lines indicate the stimulus beginning and end. (H) Percentage of responding Pbg (green) and pulvinar (orange) units for six tested visual stimuli. The dashed lines correspond to 100%, that is the total number of light responsive units (n = 70 pulvinar; n = 115 Pbg). **p<0.01 two proportion z-test. (I–K) Direction-selectivity was measured with a big-fast black square moving in eight directions. Pbg example unit (I) responding preferentially to a stimulus moving to the front and to stimuli moving to the back/down. Pulvinar example unit (J) without direction preference. Distribution of direction-selectivity indices (DSI) and two example cells with a DSI around the population average (K). *p<0.05 Kolmogorov-Smirnov test. (L–M) Median ± octiles of responses from Pbg (L) and pulvinar (M) recordings to an expanding disc. Pbg: n = 84; LP: n = 34. (N) Cumulative distributions are shown for response amplitude during the expansion. See also *Figure 6—figure supplement 1*.

The online version of this article includes the following figure supplement(s) for figure 6:

**Figure supplement 1.** Parabigeminal and pulvinar responses.

## Ganglion cell types innervating the colliculo-parabigeminal and colliculo-pulvinar circuits

The identification of the ganglion cells innervating the colliculo-parabigeminal and colliculo-pulvinar circuit was accomplished by finding the best match of each ganglion cell in our data to the cell types in the Eyewire data base using a combination of morphological and molecular cues (*Table 1*). Of the 14 clusters, 8 (clusters 1, 2, 4, 6, 9, 11, 12 and 13) include molecularly identified cells (*Figures 2* and *3*). Briefly, the cells in cluster 4 are ON-OFF direction-selective cells, based on their characteristic co-stratification with the ChAT bands and positive CART labelling (*Dhande et al., 2013*; *Sanes and Masland, 2015*). The four alpha ganglion cell types (cluster 13, sustained ON-alpha; cluster 11, transient ON-alpha; cluster 6, transient OFF-alpha; cluster 1, sustained OFF-alpha) were positively identified based on a combination of positive SMI32 staining, dendritic anatomy and large cell body size (*Bleckert et al., 2014*; *Krieger et al., 2017*), as well as their visual response properties (*Baden et al., 2016*). In addition, three out of the four FOXP2-positive cell types were identified (cluster 2, Fmini-OFF; cluster 9, Fmini-ON; cluster 12, Fmidi-ON), where Fmini-OFF cells are likely PV7 cells (*Farrow et al., 2013*; *Rousso et al., 2016*).

For the remaining six cell types (clusters 3, 5, 7, 8 10 and 14), we manually inspected published collections of anatomical and functional retinal ganglion cell types (*Baden et al., 2016*; *Sümbül et al., 2014a*; *Völgyi et al., 2009*). These included the small cells in cluster 8 that resemble the HD1 or HD2 cells (*Bae et al., 2018*; *Jacoby and Schwartz, 2017*). The large ON-cells in cluster 14 might correspond to type G6 (*Völgyi et al., 2009*), and resemble the Ka-cells (*Sümbül et al., 2014a*). Based on their size and the reported responses to a moving bar (*Bae et al., 2018*), they best fit the ON sustained (G22) or ON local sustained (G30) type (*Baden et al., 2016*). We found the best morphological match for cluster 3 and cluster 10 to be G16 and G8, respectively (*Völgyi et al., 2009*), and the best functional match to be G26 (ON DS sustained) and G17 (ON local transient) (*Baden et al., 2016*). Finally, the chirp response profiles of cluster 5 and 7 best fit the chirp responses of the mini OFF-transient cells (G9) and OFF-slow cells (G4), respectively (*Baden et al., 2016*). Although our identification of cell types is well grounded, the relationship between anatomical data sets (*Bae et al., 2018*), physiological data sets (*Baden et al., 2016*) and molecular identity of cell types remains incomplete (*Dhande et al., 2015*; *Sanes and Masland, 2015*).

## Retrograde transsynaptic labelling of retinal ganglion cells

Transsynaptic rabies tracing using injections of herpes-simplex virus (HSV-rabiesG-TVA-mCherry) to target nuclei, and subsequent injection of EnvA-coated rabies virus (EnvA-SADΔG-GCaMP6s) to the superior colliculus proved to be a suitable tool to determine the circuit specificity of collicular projecting retinal ganglion cells. First, HSV has a strictly synaptic uptake mechanism that prevents infection of passing axons, ensuring that we labeled neurons that synapse within the pulvinar or parabigeminal nucleus (*Antinone and Smith, 2010*; *McGavern and Kang, 2011*). Labeling of passing axons is an issue with other retrograde tracers (*Ellis et al., 2016*). Second, G-deleted rabies has been demonstrated to reliably label retinal ganglion cells innervating the superior colliculus, dorsal lateral geniculate nucleus or medial terminal nucleus (*Cruz-Martín et al., 2014*; *Ellis et al., 2016*; *Farrow et al.,*

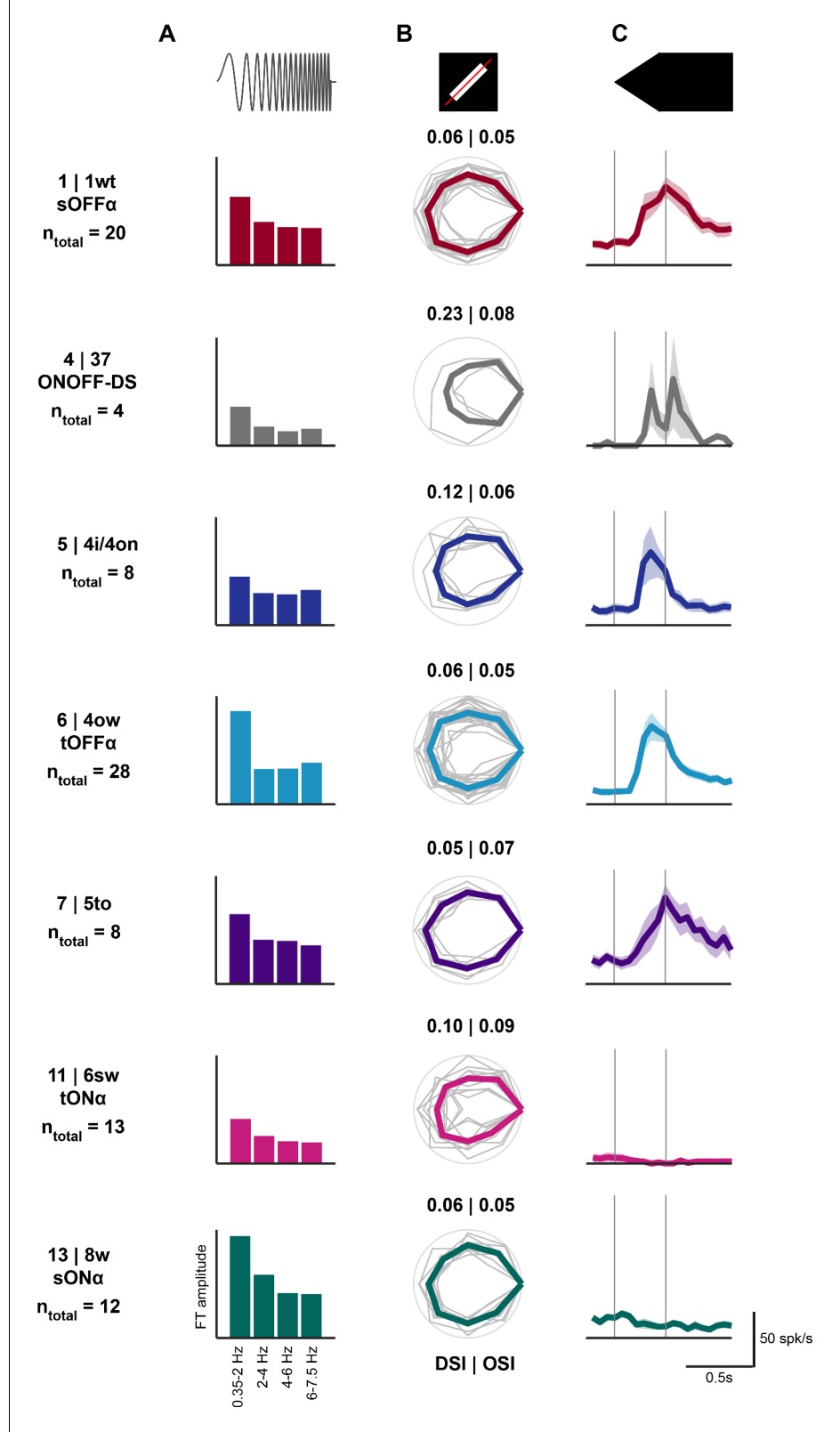

**Figure 7.** Visual responses of retinal ganglion cells innervating the colliculo-parabigeminal and the colliculo-pulvinar circuits. $n_{total}$ indicates the number of cells with patch recording. (A) Mean Fourier Transform amplitude for four different frequency bands. (B) Peak response to each of the 8 directions of the moving bar. Peak responses were normalized for each cell to its maximum and the polar plot was rotated so that the maximal

*Figure 7 continued*
response is on the right. Gray: individual cells; color: mean polar plot. Direction-selectivity index (DSI) and orientation-selectivity indices (OSI) are indicated. (C) Mean ± SEM of responses to a linearly expanding disc.

*2013*; *Rompani et al., 2017*; *Yonehara et al., 2013*). Finally, injection of EnvA-coated rabies virus into the superior colliculus, without previous injection of HSV-rabiesG-TVA-mCherry into either the pulvinar or parabigeminal nucleus resulted in no labeling of retinal ganglion cells in the retina or neurons in the superior colliculus (*Figure 1—figure supplement 1*). We therefore believe that this combination of tools reliably and specifically labels retinal ganglion cells innervating the different targeted pathways of the superior colliculus.

However, while there is no evidence suggesting a retinal ganglion cell bias in rabies virus uptake, the speed with which viral particles are retrogradely transported likely varies between retinal ganglion cell types, due to differences in axonal diameter and the availability of minus-end-directed motor dynein of different cell types (*Antinone and Smith, 2010*). These biases are reflected in the relatively large numbers of alpha retinal ganglion cells we labeled as compared to the number of small retinal ganglion cells (e.g. ON-OFF direction-selective and FOXP2$^+$ ganglion cells) labeled in our individual experiments (*Figure 3—figure supplement 1*).

In addition, it is possible that we underestimated the number of cell types innervating these two circuits. This is for three reasons. First, clusters with < 1% of all cells were not considered as an input-providing cell type here (*Figure 2—source data 1*). If these clusters are 'true' inputs, a higher infection rate might reveal enough cells to be considered for further analysis. Second, we saw a large variability in the absolute numbers of neurons labeled in different experiments (*Figure 3—figure supplement 1*). For cell types with a low probability of being labeled they may not have been detected reliably enough to be counted. However, within each circuit, we measured a similar percentage of molecularly identified cells independent of the total number of rabies-infected cells (*Figure 3—figure supplement 1*). Third, we systematically labeled more neurons in pulvinar experiments as compared to parabigeminal experiments. This might be because the parabigeminal nucleus is difficult to target due to its small size or because the pulvinar receives more inputs from the colliculus. Despite this fact, four cell types (clusters 3, 4, 7 and 14) were found almost exclusively in our parabigeminal experiments.

Given these potential biases due to technical limitations, we have more cells from colliculo-pulvinar experiments in our database than from colliculo-parabigeminal experiments. The absolute number of cells found in a given cluster for each circuit is a consequence of these experimental limitations and not of the innervation strength of this cell type. We therefore do not assess the relative input strength of the different ganglion cell types to an individual circuit. Instead, we based our analysis and conclusions on a comparison of the relative distributions of individual cell types between the two circuits, and not absolute numbers. By comparing the relative percentage of the same cell types between the circuits, we have minimized this effect. We are confident that the differences in numbers of infected cells has no major effect on the relative distributions of cell types observed, and the differences we see in labeling probabilities reflect real biological differences in the wiring diagram of the two circuits. To get a complete picture of how individual neurons in each circuit are sampling retinal inputs, a single cell or sparse cell-type-specific approach is necessary (*Rompani et al., 2017*; *Yonehara et al., 2013*).

A second tool used to limit infecting off-target brain nuclei during the injection of HSV was the *Ntsr1-GN209Cre* mouse line (*Gerfen et al., 2013*). This mouse line ensured that we exclusively labeled wide-field neurons projecting to the pulvinar (*Gale and Murphy, 2014*). This may bias our results as it is possible that unknown collections of other cell types also project to the pulvinar from the superior colliculus. However, two pieces of evidence suggest this is unlikely. First, in a screen of different Cre-mouse lines, *Ntsr1-GN209Cre* positive neurons were found to only innervate the pulvinar, while other cell types were found to not innervate the pulvinar (*Gale and Murphy, 2018*; *Gale and Murphy, 2014*). In addition, unbiased retrograde labeling of collicular neurons, using HSV, from the pulvinar has predominantly revealed wide-field neurons, though a small number of neurons that might be of a different type were also seen (*Zhou et al., 2017*). *Shang et al. (2018)* report that a subpopulation of neurons in the PV-Cre mouse line projects to the posterior portion of the pulvinar. As these pulvinar projecting PV$^+$ neurons have a similar projection pattern and cell body

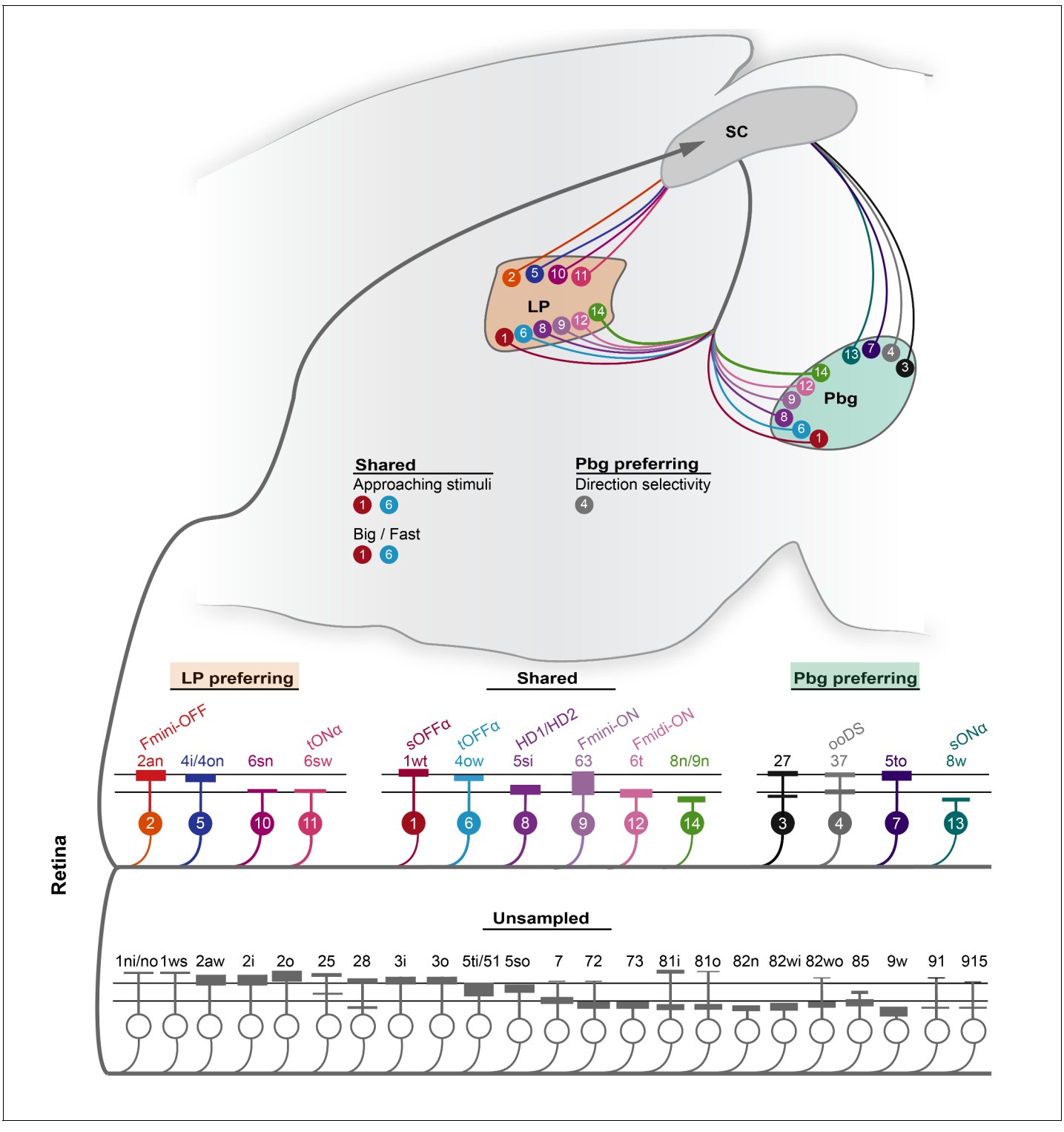

**Figure 8.** Schematic of the projection preference logic of retinal ganglion cell inputs to the superior colliculus. In total, 14 ganglion cell types are sampled by parabigeminal- and pulvinar-projecting collicular neurons. Four cell types are biased for the LP-circuit, four have a preference for the Pbg-circuit, and six have no bias for either circuit. The color and the number of the retinal ganglion cell types corresponds to the clusters defined in *Figure 2*. All the other cell types of the EM data set are not sampled by our data set. The relative response properties routed to the pulvinar and/or parabigeminal nucleus are indicated with the relative color and number.

**Table 1.** Overview of the retinal inputs to the colliculo-parabigeminal and colliculo-pulvinar circuits. Top: Average stratification profile for each cluster. In bold are associations with published retinal ganglion cell types that are supported by molecular markers and the assignments in *Bae et al. (2018)*. The best matches defined by visual inspection of the published anatomical and functional ganglion cell libraries are listed in normal font.

| | | 3 | 4 | 7 | 13 | 1 | 6 | 8 | 9 | 12 | 14 | 2 | 5 | 10 | 11 |
|---|---|---|---|---|---|---|---|---|---|---|---|---|---|---|---|
| **Anatomy** | | Parabigeminal nucleus | | | | Shared | | | | | | Pulvinar | | | |
| | Cluster # | 3 | 4 | 7 | 13 | 1 | 6 | 8 | 9 | 12 | 14 | 2 | 5 | 10 | 11 |
| | EM name | 27 | 37 | 5to | 8w | 1wt | 4ow | 5si | 63 | 6t | 8n/9n | 2an | 4i/4on | 6sn | 6sw |
| | Marker | | CART | | SMI32 | SMI32 | SMI32 | | FOXP2 | FOXP2 | | FOXP2 | | | SMI32 |
| | Putative type | | onoff-DS | | sONa | sOFFa | tOFFa | HD1/HD2 | Fmini-ON | Fmidi-ON | | Fmini-OFF | | | tONa |
| **Putative anatomical types** | Sümbül et al. 2014 | | D | | | I | F | | | | B | H | | | |
| | Völgyi et al. 2009 | G16 | G20/22 | | G2 | G3 | G4/5 | G14 | G18 | | G6 | G18 | | G8 | G9 |
| | Krieger et al. 2017 | | | | sONa | sOFFa | tOFFa | | | | | | | | tONa |
| | Rousso et al. 2018 | | | | | | | | Fmini-ON | Fmidi-ON | | Fmini-OFF | | | |
| | Jacoby et al. 2017 | | | | | | | HD1, HD2 | | | | | | | |
| | Farrow et al. 2013 | | PV0 | | PV1 | PV6 | PV5 | | | | | PV7 | | | |
| | Baden et al. 2016 | G26 | **G12** | G4 | **G24** | G5 | G8 | G11/14 | G18 | G15/20 | G22/30 | G2 | G9 | G17 | **G19** |
| **Known functional properties** | | ON responses | Direction-selective; small objects; expansion responses | Sustained OFF; big objects | Big, fast objects | Big, fast objects | 'looming detector', preference for big objects | ON-OFF responses; small objects; 'high-definition' | Small, rather slow objects | Small, rather slow objects | Very sustained ON-responses | Small objects | Expansion response (local stimuli) | Strong responses to big and small objects | |

position within the superior colliculus to *Ntsr1-GN209Cre* neurons, we think it likely that they are a subpopulation of wide-field neurons. However, while in mice only one wide-field neuronal cell type has been described, two types of wide-field neurons have been found in ground squirrels that have distinct anatomies and project to different regions within the pulvinar (*Fredes et al., 2012*; *Major et al., 2000*). The description of cell types within the mouse superior colliculus remains immature compared to that of mouse retinal ganglion cell types.

## Functional responses of retinal ganglion cells and target nuclei

One question we attempted to answer here was whether we could explain the different visual response properties of neurons in the parabigeminal nucleus and pulvinar by their distinct retinal inputs. While some of the differences were indeed consistent (e.g. direction-selective neurons innervate the colliculo-parabigeminal circuit and looming sensitive neurons innervate both circuits), we found clear differences in the visual responses in the retina and the pulvinar and parabigeminal nucleus. The classification of retinal ganglion cells was based on their generally robust responses to a full-field chirp stimulus, which fails to evoke visual responses in either the posterior pulvinar or parabigeminal nucleus (*Figure 6—figure supplement 1*). In addition, the colliculo-pulvinar circuit receives inputs from ganglion cells that respond well to big and fast objects (cluster 11 and shared inputs from cluster 1), but responses to such stimuli were weak or absent in the pulvinar neurons. These differences might reflect strong non-linearities in how retinal inputs are integrated, or they might be a result of balanced excitatory and inhibitory inputs that cancel each other out.

These differences between visual response properties of innervating retinal ganglion cells and their disynaptic central targets in the pulvinar and parabigeminal nucleus is in stark contrast to what is found in the dorsal lateral geniculate nucleus, where many of the neurons respond well to full-field stimulation and their visual response properties can be understood as a linear sum of different combinations of putatively innervating retinal ganglion cell types (*Roson et al., 2019*). Local inhibitory and excitatory connections within the superior colliculus might mediate the different full-field responses of inputs and outputs (*Gale and Murphy, 2018*). Indeed, while inputs from the visual cortex appear to modulate the gain of visual responses within the superior colliculus (*Shi et al., 2017*; *Wang et al., 2010*; *Zhao et al., 2014*; *De Franceschi and Solomon, 2018*), removal of local inhibition in the superior colliculus reveals masked response characteristics including responses to large, stationary objects in pulvinar-projecting neurons (*Gale and Murphy, 2016*). A more in-depth, cell-type-specific approach is needed to understand the functional consequences of selective wiring of retinal ganglion cells with their targets in the superior colliculus.

We recorded direction-selective responses in the parabigeminal nucleus but not in the pulvinar, which was mirrored by the selective innervation of ON-OFF direction-selective neurons to the colliculo-parabigeminal pathway. However, Fmini-ON, which innervate both circuits, and Fmini-OFF cells, which selectively innervate the colliculo-pulvinar circuit, have been reported to be direction-selective (*Rousso et al., 2016*). There are three reasons why this selectivity may not make a major contribution to direction-selective responses in the superior colliculus and its downstream targets. First, it has been demonstrated that direction-selective responses in the superior colliculus rely on the inhibitory output of starburst amacrine cells (*Shi et al., 2017*), where starburst amacrine cells are responsible for the direction-selective responses of ON-OFF and ON direction-selective ganglion cells (*Euler et al., 2002*; *Fried et al., 2002*; *Hillier et al., 2017*; *Pei et al., 2015*; *Yonehara et al., 2016*; *Yonehara et al., 2013*; *Yoshida et al., 2001*). However, starburst amacrine cells have not been implicated in mediating the direction-selective responses of highly asymmetric retinal ganglion cell types including the Fmini and JAM-B neurons (*Joesch and Meister, 2016*; *Kim et al., 2008*; *Rousso et al., 2016*). Second, unlike ON-OFF direction-selective neurons, the direction-selectivity of Fmini neurons is highly speed dependent, with a peak selectivity at 585 µm/s and negligible selectivity at speeds greater than 1300 µm/s (*Rousso et al., 2016*). Direction-selective responses in the parabigeminal nucleus were recorded at speeds equivalent to more than 1500 µm/s on the retina. We did not observe direction-selective responses in the pulvinar at speeds where Fmini neurons are direction-selective (*Figure 6* and *Figure 6—figure supplement 1*). It is therefore unlikely that the Fmini neurons are contributing to these direction-selective responses. Finally, like Fmini neurons, the asymmetric JAM-B neurons were originally identified as direction-selective, however, unlike ON-OFF direction-selective neurons, their direction-selectivity is not a robust property. *Kim et al. (2008)* reported that the direction-selectivity of JAM-B neurons is highly dependent on each neuron's

individual dendritic asymmetry, while it has been reported that the direction-selective, but not orientation-selective, responses of JAM-B cells are sensitive to light conditions (*Joesch and Meister, 2016*; *Nath and Schwartz, 2017*). The many similarities between Fmini and JAM-B neurons suggest that a more extensive exploration of their response properties is necessary before they are determined to be robust encoders of directional information in the visual scene (*Rousso et al., 2016*).

The two neural circuits investigated here are each known to mediate visually guided aversive behaviors (*Shang et al., 2018*; *Shang et al., 2015*; *Wei et al., 2015*). In this context, the responses to biologically relevant stimuli of the ganglion cells innervating the two circuits are of interest. We found that neurons in the pulvinar respond poorly to large stimuli, but responded to small, slowly moving stimuli, which have been suggested to mimic a distant predator (*Zhang et al., 2012*). In addition, neurons in the pulvinar and parabigeminal nucleus respond well to quickly expanding dark stimuli, which are thought to mimic a quickly approaching threat (*De Franceschi et al., 2016*; *Dean et al., 1989*; *Yilmaz and Meister, 2013*). While robust responses to approaching stimuli have been reported in both pulvinar-projecting and parabigeminal-projecting collicular neurons, only pulvinar-projecting collicular neurons have been reported to respond to small slowly moving stimuli (*Beltramo and Scanziani, 2019*; *Bennett et al., 2019*; *Gale and Murphy, 2016*; *Gale and Murphy, 2014*; *Shang et al., 2018*; *Shang et al., 2015*; *Maaten and Hinton, 2008*; *Inayat et al., 2015*). Consistent with these results, we found that the putative ganglion cell types preferentially sampled by the colliculo-pulvinar circuit have smaller dendritic fields, including the Fmini-OFF cells (cluster 2), which respond to small, dark stimuli. In addition, the pulvinar receives strong input from cluster 10, which based on the small dendritic size and transient responses to a moving bar (*Bae et al., 2018*) could correspond to the local ON-cells (G17) (*Baden et al., 2016*) and hence provide information about local movement. Transient OFF-alpha cells (cluster 6) are known to preferentially respond to expanding stimuli and could mediate these responses in both circuits (*Shang et al., 2018*; *Shang et al., 2015*; *Wei et al., 2015*). Retinal ganglion cells with a bias for the colliculo-parabigeminal circuit have larger dendritic fields and their putative function is to respond to large moving objects and their motion direction (clusters 4, 7 and 13), and we found similar stimulus preferences in the parabigeminal neurons. Together the parabigeminal-preferring ganglion cells might detect predators attacking from angles that are not recognized by expansion-detectors.

Determining the link between the visual responses of retinal ganglion cells, their central brain targets and behavior remains a central question in visual neuroscience (*Hillier et al., 2017*; *Hubel and Wiesel, 1961*; *Lettvin et al., 1959*; *Liang et al., 2018*; *Roson et al., 2019*; *Temizer et al., 2015*). The approaches used to achieve this have predominantly involved recording responses in the retina independent of recording in the brain, or during behavior (*Hillier et al., 2017*; *Lettvin et al., 1959*; *Usrey et al., 1998*). While attempts to link particular cell types in the retina with responses in central brain structures and behavior have been undertaken, clear mechanistic relationships remain limited (*Chen et al., 2011*; *Hillier et al., 2017*; *Liang et al., 2018*; *Roson et al., 2019*; *Shi et al., 2017*; *Yonehara et al., 2016*). We believe development of approaches that enable simultaneous recordings of identified retinal cell types and central brain activity in awake behaving animals will allow us to better understand these relationships (*de Malmazet et al., 2018*; *Hong et al., 2018*; *Liang et al., 2018*).

## Visual pathways through the superior colliculus

Studies investigating the organization of retinal inputs to single cells in the lateral geniculate nucleus have suggested that there is a large degree of fuzziness/variability in the information each neuron receives from the retina (*Hammer et al., 2015*; *Liang et al., 2018*; *Morgan et al., 2016*; *Rompani et al., 2017*; *Roson et al., 2019*). Here, we demonstrate that in the superior colliculus a high degree of regularity exists if one considers the projection targets. This data suggests there are strict limits on the degree of mixing of retinal ganglion cell inputs that occurs in the superior colliculus, where each output pathway has access to a distinct, only partially overlapping, set of visual information encoded by the retina. The observed regularity could exist either because the superior colliculus has a more 'hard-wired' architecture; or because we focused on projection-specific disynaptic circuits. When considering the layer-specific targets of the lateral geniculate nucleus in the visual cortex, Cruz-Martin et al. suggest that direction-selective neurons are preferentially sampled by layer one projecting neurons of the thalamus (*Cruz-Martín et al., 2014*). We propose that understanding the specific input structure to neurons and cell types with different projection profiles will

greatly enhance our ability to create mechanistic models of how information from the sensory periphery informs the triggering of behaviors and decision making.

# Materials and methods

## Key resources table

| Reagent (type) or Resource | Designation | Source or reference | Identifiers | Additional information |
|---|---|---|---|---|
| Strain, strain background (*Mus musculus*) | C57BL/6 | Jackson laboratory | JAX:000664 | |
| Genetic reagent (*Mus musculus*) | PV-Cre (B6;129P2-Pvalb$^{tm1(cre)Arbr}$/J) | Jackson laboratory | JAX:008069 | |
| Genetic reagent (*Mus musculus*) | Ai9 (B6.Cg-Gt(ROSA)26Sor$^{tm9(CAG-tdTomato)Hze}$/J) | The Jackson laboratory | JAX:007909 | |
| Genetic reagent (*Mus musculus*) | Gad2-IRES-CRE | The Jackson laboratory | JAX: 10802 | |
| Genetic reagent (*Mus musculus*) | Tg(Ntsr1-cre)GN209 Gsat/Mmucd | Laboratory of Keisuke Yonehara | RRID:MMRR_030780-UCD | |
| Antibody | anti-GFP (rabbit polyclonal) | Thermo Fisher Scientific | Cat# A-11122; RRID:AB_221569 | 1:500 |
| Antibody | anti-GFP (chicken, polyclonal) | Thermo Fisher Scientific | Cat# A-10262; RRID:AB_2534023 | 1:500 |
| Antibody | anti-ChAT (goat, polyclonal) | Millipore | Cat# AB144P RRID:AB_11214092 | 1:200 |
| Antibody | SMI32 (mouse, monoclonal) | Biolend | Cat# 801701; RRID:AB_2564642 | 1:1000 |
| Antibody | anti-CART (rabbit, polyclonal) | Phoenix Pharmaceuticals | H-003–62; RRID:AB_2313614 | 1:500 |
| Antibody | anti-FOXP2 (goat, polyclonal) | abcam | Cat# 1307; RRID:AB_1268914 | 1:2000 |
| Antibody | anti-mCherry (chicken, polyclonal) | Novus | Cat# NBP2-25158 RRID:AB_2636881 | 1:1000 |
| Antibody | Alexa 488 donkey anti-rabbit | Thermo Fisher Scientific | Cat# A-21206; RRID:AB_2535792 | 1:500–1000 |
| Antibody | Alexa 488 donkey anti-chicken | Immuno-Jackson | Cat# 703-545-155 RRID:AB_2340375 | 1:500 |
| Antibody | Alexa 633 donkey anti-goat | Thermo Fisher Scientific | Cat# A-21082 RRID:AB_10562400 | 1:500 |
| Antibody | Cy3 donkey anti-mouse | Immuno-Jackson | Cat# 715-165-151 RRID:AB_2315777 | 1:400 |
| Antibody | Alexa 555 donkey anti-goat | abcam | Cat# ab150130 | 1:300 |
| Antibody | DyLight 405 donkey anti-rabbit | Immuno-Jackson | Cat# 715-475-150 RRID:AB_2340839 | 1:200 |
| Antibody | Cy3 donkey anti-chicken | Immuno-Jackson | Cat# 703-166-155 RRID:AB_2340364 | 1:800–1000 |
| Antibody | 435/455 Nissl Stain | Thermo Fisher Scientific | Cat# N21479 | 1:150 |
| Antibody | DAPI | Roche | Cat# 10276236001 | 1:1000 |
| Peptide, recombinant protein | Cholera Toxin Subunit B conjugated with Alexa488 | Thermo Fisher Scientific | Cat# C22841 | |
| Chemical compound, drug | Lipophilic tracers DiI, DiD, DiO | Thermo Fisher Scientific | Cat# D7776, D7757, D275 | |

*Continued on next page*

*Continued*

| Reagent (type) or Resource | Designation | Source or reference | Identifiers | Additional information |
|---|---|---|---|---|
| Strain, strain background (SAD-B19 rabies virus) | Rabies virus: G-coated SAD-△G-GCaMP6s | This paper | N/A | Is available upon request or from Laboratory of Botond Roska |
| Strain, strain background (SAD-B19 rabies virus) | Rabies virus: EnvA-coated SAD-△G-GCaMP6s | This paper | N/A | Is available upon request or from Laboratory of Botond Roska |
| Strain, strain background (herpes simplex virus) | HSV: hEF1a-TVA950-T2A-RabiesG-IRES-mCherry | MIT core | RN714 | |
| Strain, strain background (herpes simplex virus) | HSV: hEF1a-LS1L-TVA950-T2A-Rabies G-IRES-mCherry | MIT core | RN716 | |
| Chemical compound, drug | 10x PBS | VWR | Cat# 437117K | |
| Chemical compound, drug | 1x PBS | VWR | Cat# 444057Y | |
| Chemical compound, drug | Histofix 4% | Roche | Cat# P087.5 | |
| Chemical compound, drug | Normal Donkey Serum | Millipore | Cat# 30–100 ML | |
| Chemical compound, drug | 10% Bovine Albumin | Sigma | Cat# SRE0036-250ML | |
| Reagent | DABCO | Sigma | Cat# 290734 | |
| Chemical compound, drug | DMEM, high-glucose | Thermo Fisher Scientific | Cat# 41965062 | |
| Chemical compound, drug | Trypsin 0.05% | Thermo Fisher Scientific | Cat# 25300054 | |
| Chemical compound, drug | Fetal Bovine Serum (FBS) | Thermo Fisher Scientific | Cat# 10270106 | |
| Chemical compound, drug | 2,2'-thiodiethanol (TDE) | Sigma | Cat# 166782–500G | |
| Chemical compound, drug | ProLong Gold Antifade Mounting Medium | Thermo Fisher Scientific | Cat# P36934 | |
| Chemical compound, drug | Sodium Azide ($NaN_3$) | Sigma | Cat# S2002-100G | |
| Chemical compound, drug | Triton X-100 | Sigma | Cat# S8875 | |
| Chemical compound, drug | Sodium Chloride (NaCl) | Sigma | Cat# S7653-250G | |
| Chemical compound, drug | Potassium Chloride (KCl) | Sigma | Cat# P5405-25G | |
| Chemical compound, drug | Calcium Chloride ($CaCl_2$) | Sigma | Cat# C5670-100G | |
| Chemical compound, drug | Magnesium Chloride ($MgCl_2$) | Sigma | Cat# 4880 | |
| Chemical compound, drug | D-glucose (Dextrose) | Sigma | Cat# D9434-250G | |
| Chemical compound, drug | Sodium phosphate monobasic ($NaH_2PO_4$) | Sigma | Cat# S5011 | |

*Continued on next page*

*Continued*

| Reagent (type) or Resource | Designation | Source or reference | Identifiers | Additional information |
|---|---|---|---|---|
| Chemical compound, drug | Sodium Hydroxide (NaOH) | Sigma | Cat# 655104–500G | |
| Chemical compound, drug | Sodium bicarbonate (NaHCO$_3$) | Sigma | Cat# S8875-1KG | |
| Cell Line (Mesocricetus auratus) | BHK cells | Laboratory of Botond Roska/Laboratory of Karl-Klaus Conzelmann | N/A | |
| Cell Line (Mesocricetus auratus) | B7GG cells | Laboratory of Botond Roska/Laboratory of Karl-Klaus Conzelmann | N/A | |
| Cell Line (Mesocricetus auratus) | BHK-EnvA cells | Laboratory of Botond Roska/Laboratory of Karl-Klaus Conzelmann | N/A | |
| Cell Line (*Homo sapiens*) | HEK293T-TVA cells | Laboratory of Botond Roska/Laboratory of Karl-Klaus Conzelmann | N/A | |
| Software, algorithm | Fiji | *Schindelin et al. (2012)* | RRID:SCR_002285 | |
| Software, algorithm | MATLAB | Mathworks | RRID:SCR_001622 | |
| Software, algorithm | Zen lite | Zeiss | | |
| Software, algorithm | CAFFE | | caffe.berkeleyvision.org | |
| Software, algorithm | ChAT band detector | This paper | https://github.com/farrowlab/ChATbandsDetection | |
| Software, algorithm | VNET | | github.com/faustomilletari/VNet | |
| Software, algorithm | PYTHON | Python Software Foundation | www.python.org | |
| Software, algorithm | t-distributed Stochastic Neighbor Embedding | *Maaten and Hinton, 2008* | | |
| Software, algorithm | CANDLE | *Coupé et al., 2012* | | |
| Software, algorithm | sparse PCA | | http://www2.imm.dtu.dk/projects/spasm | |
| Software, algorithm | nanconv | Benjamin Kraus | http:// mathworks.com/matlabcentral/fileexchange/41961-nanconv | |
| Software, algorithm | retistruct | *Sterratt et al., 2013* | http://davidcsterratt.github.io/retistruct/ | |
| Software, algorithm | SpikeGLX | | https://billkarsh.github.io/SpikeGLX/ | |
| Software, algorithm | GNU Octave | Free Software Foundation | www.gnu.org/software/octave | |
| Software, algorithm | Psychophysics Toolbox | Psychtoolbox | http://psychtoolbox.org | |
| Software, algorithm | SpyKING CIRCUS | Yger et al. 2018 | https://spyking-circus.readthedocs.io | |

*Continued on next page*

*Continued*

| Reagent (type) or Resource | Designation | Source or reference | Identifiers | Additional information |
|---|---|---|---|---|
| Software, algorithm | Phy | Cortex Lab at University College London | https://phy-contrib.readthedocs.io https://github.com/kwikteam/phy | |
| Software, algorithm | WaveSurfer (version: 0.918) | Janelia Research Campus | http://wavesurfer.janelia.org/ | |
| Software, algorithm | ScanImage | Vidrio Technoloies | http://scanimage.vidriotechnologies.com | |
| Software, algorithm | Allen CCF Tools | *Shamash et al. (2018)* | https://github.com/cortex-lab/allenCCF | |
| Software, algorithm | TREES toolbox | *Cuntz et al. (2011)* | https://github.com/cuntzlab/treestoolbox | |
| Other | Rapid Flow Filters 0.2 µm pore size | VWR | Cat# 514–0027 | |
| Other | Premium Standard Wall Borosilicate capillary glass | Warner Instrument | Cat# G100-4 | |
| Other | Wiretrol II capillary micropipettes | Drumond Scientific | Cat# 5-000-2005 | |
| Other | Borosilicate glass | Sutter Instrument | Cat# BF100-20-10 | |
| Other | Laser-Based Micropipette Puller | Sutter Instrument | Cat# P-2000 | |
| Other | Small Animal Stereotaxic Workstation | Narishige | Cat# SR-5N | |
| Other | Stereotaxic Micromanipulator | Narishige | Cat# SM-15R | |
| Other | Hydraulic Oil Micromanipulator | Narishige | Cat# MO-10 | |
| Other | Oil Microinjector | Narishige | Cat# IM-9B | |
| Other | Two-photon microscope | Scientifica | Serial# 14200 | |
| Other | 780 nm LED light source | Thorlabs | Cat# M780L3 | |
| Other | Patch-Clamp amplifier | Molecular Device | Axon Multiclamp 700B | |
| Other | Patch-Clamp microscope | Scientifica | Slice Scope | |
| | Patch-Clamp manipulator | Scientifica | Serial# 301311 | |
| Other | Zeiss LSM 710 confocal microscope | Zeiss | Cat# LSM710 | |
| Other | Neuropixels phase 3A system | Imec | | |
| Other | FPGA Kintex-7 KC705 | Xilinx | EK-K7-KC705-G | |
| Other | Micromanipulator | Sensapex | Cat# uMp-1 | |

## Experimental model and subject details

In total, 97 mice (3–5 weeks old for virus injections, 2–3 months for in vivo physiology) of either sex were used in our experiments including *PvalbCre*, *PvalbCre x Ai9*, *Ntsr1-GN209Cre*, *Ntsr1-GN209Cre x Ai9*, and *Gad2Cre*. *PvalbCre* mice (JAX: 008069) (*Hippenmeyer et al., 2005*) express Cre recombinase in parvalbumin-expressing neurons. *Ntsr1-GN209Cre* mice (Genset: 030780-UCD) express Cre recombinase in *Ntsr1-GN209*-expressing neurons. *Gad2Cre* mice (JAX: 010802) express Cre recombinase in *Gad2*-expressing neurons. Ai9 (JAX: 007909) is a tdTomato reporter mouse line (*Madisen et al., 2010*). Animals were maintained on a 12 hr light/dark cycle, and fed with sterilized

food, water, bedding and nesting material. All animal procedures were performed in accordance with standard ethical guidelines of KU Leuven and European Communities Guidelines on the Care and Use of Laboratory Animals (004–2014/EEC, 240–2013/EEC, 252–2015/EEC).

## Method details

### Rabies virus production

Rabies production method was similar to previously published methods (*Osakada and Callaway, 2013*; *Yonehara et al., 2013*). Glycoprotein G-coated, G-deleted B19 rabies virus (G-coated SAD-ΔG-GCaMP6s RV) was amplified in B7GG cells, which express rabies glycoprotein G. For amplification, approximately $10^6$ infectious units of G-coated SAD-ΔG-GCaMP6s RV were used to infect five 10 cm plates of 80% confluent B7GG cells followed by 2–6 hr of incubation. Then, infected B7GG cells were treated with 0.05% trypsin (Thermo, 25300054) and split into twenty-five 10 cm plates. To harvest the virus, we collected the supernatant of the infected cells every 3 days. 5–6 harvests were performed. To concentrate the virus, the supernatant was firstly centrifuged at 2500 RPM and filtered (VWR, 514–0027) to get rid of the cell debris. Then the virus was spun in an ultracentrifuge for 5–12 hr at 25,000 RPM and at 4˚C. After ultracentrifugation, the supernatant was discarded, and the pellet was dissolved in 200 µl of the original cell culture supernatant. The virus was tittered by counting a culture of infected BHK cells. To produce EnvA-coated SAD-ΔG-GCaMP6s RV, approximately $10^6$ infectious units of G-coated SAD-ΔG-GCaMP6s RV were used to infect BHK-EnvA cells. The same procedure as for the G-coated RV amplification was then applied. EnvA-coated SAD-ΔG-GCaMP6s RV was tittered by infection of HEK293T-TVA cells. The titer used for injection ranged from $10^7$ to $10^9$ infectious units/ml (IU/ml).

### Surgical procedures

Animals were quickly anesthetized with Isoflurane (Iso-vet 1000 mg/ml) and then injected with a mixture of Ketamine and Medetomidine (0.75 mL Ketamine (100 mg/mL) + 1 mL Medetomidine (1 mg/mL) + 8.2 mL Saline). Mice were placed in a stereotaxic workstation (Narishige, SR-5N). Dura tear (NOVARTIS, 288/28062–7) was applied to protect the eyes. To label the ganglion cells in the parabigeminal nucleus circuit, we performed the surgery on wild type mice and injected herpes-simplex-virus (HSV, hEF1a-TVA950-T2A-rabiesG-IRES-mCherry, MIT viral core, RN714) and EnvA-coated SAD-ΔG-GCaMP6s RV. In our experiment, we used *PV-Cre* mice as wild type mice. For the first injection of HSV into the parabigeminal nucleus, we used micropipettes (Wiretrol II capillary micropipettes, Drumond Scientific, 5-000-2005) with an open tip of around 30 µm and an oil-based hydraulic micromanipulator MO-10 (Narishige) for stereotactic injections. Alternatively, we used an oil-based microinjector IM-9B (Narishige) with the corresponding micropipettes (Warner Instrument, G100-4) with an open tip of 30 µm. The injection coordinates for a 4 weeks old mouse with a bregma-lambda distance of 4.7 mm were AP: −4.20; ML:±1.95; DV: 3.50 mm. As the mice were different in body size, we adjusted the coordinates for each mouse according to their bregma-lambda distance. To label the injection sites, DiD (Thermo, D7757) was used to coat the pipette tip. We injected in total 100–400 nl HSV in single doses of up to 200 nl with a waiting time of 5–10 min after each injection. Twenty-one days later, we injected rabies virus (EnvA-coated SAD-ΔG-GCaMP6s) into the superior colliculus using the same method as for the HSV injections. The retinotopic location of the first injection into the parabigeminal nucleus or the pulvinar is unknown. To maximize the labelling of ganglion cells in the retina, we thus covered as much as possible of the superficial layer of the superior colliculus during the second injection. We injected 100–200 nl of rabies virus at a depth of 1.7–1.8 mm at four different locations within a 1 mm² field anterior of lambda and starting at the midline.

To label the pulvinar circuit, we performed the surgery on *Ntsr1-GN209Cre* mice and injected a conditional HSV (hEF1a-LS1L-TVA950-T2A-RabiesG-IRES-mCherry, MIT viral core, RN716) and EnvA-coated SAD-ΔG-GCaMP6s RV. The injections into pulvinar and superior colliculus were the same as described for the parabigeminal nucleus. The injection coordinates for the pulvinar in a 4 weeks old mouse with a bregma-lambda distance of 4.7 mm were AP: −1.85; ML:±1.50; DV: 2.50 mm.

Following injection, the wound was closed using Vetbond tissue adhesive (3M,1469). After surgery, mice were allowed to recover on top of a heating pad and were provided with soft food and water containing antibiotics (emdotrim, ecuphar, BE-V235523).

## Retina immunohistochemistry

Mouse retinas were extracted eight days after the rabies virus injection into the superior colliculus. After deep anesthesia (120 µl of Ketamine (100 mg/ml) and Xylamine (2%) in saline per 20 g body weight), eyes were gently touched with a soldering iron (Weller, BP650) to label the nasal part of the cornea and then enucleated. The retinas were extracted in 1x PBS (Diluted from 10x PBS (VWR, 437117K), pH 7.4) and three cuts were made to label the nasal, dorsal and ventral retina.

The dissected retinas were fixed in 4% paraformaldehyde (Histofix, ROTH, P087.5mm) with 100 mM sucrose for 30 min at 4°C, and then transferred to a 24-well plate filled with 1x PBS and washed three times for 10 min at room temperature or transferred into 15 ml 1x PBS and washed overnight or longer at 4°C. After washing, retinas were transferred to wells containing 10% sucrose in 1x PBS with 0.1% NaN$_3$ (w/v) and allowed to sink for a minimum of 30 min at room temperature. Then retinas were transferred to wells containing 20% sucrose in 1x PBS with 0.1% NaN$_3$ (w/v) and allowed to sink for a minimum of 1 hr at room temperature. Finally, retinas were put into 30% sucrose in 1x PBS with 0.1% NaN$_3$ (w/v) and allowed to sink overnight at 4°C. The next day, freeze-cracking was performed: retinas were frozen on a slide fully covered with 30% sucrose for 3–5 min on dry ice. The slides were then thawed at room temperature. The freeze–thaw cycle was repeated two times. Retinas were washed 3 times for 10 min each in 1x PBS, followed by incubation with blocking buffer (10% NDS, 1% BSA, 0.5% TritonX-100, 0.02% NaN$_3$ in 1x PBS) for at least 1 hr at room temperature. Primary antibody solution was added after blocking and retinas were incubated for 5–7 days under constant gentle shaking at room temperature. Primary antibodies were rabbit anti-GFP (Invitrogen, A-11122, 1:500) and goat anti-ChAT (Chemicon, Ab144P, 1:200). They were prepared in 3% NDS, 1% BSA, 0.5% TritonX-100, 0.02% NaN$_3$ in 1x PBS. After incubation, retinas were washed three times for 10 min in 1x PBS with 0.5% TritonX-100 before being transferred into the secondary antibody solution (Alexa488 donkey anti-rabbit (Invitrogen, A21206, 1:500) and Alexa633 donkey anti-goat (Invitrogen A-21082, 1:500); prepared in 3% NDS, 1% BSA, 0.5% TritonX-100, 0.02% NaN$_3$ in 1x PBS). Nuclei were stained with DAPI (Roche, 10236276001, 1:500) together with the secondary antibody solution. The retinas were incubated in the secondary antibody with DAPI solution overnight at 4°C. Retinas were then washed three times in 1x PBS with 0.5% TritonX-100 and 1 time in 1x PBS. For mounting, we used 2,2'-Thiodiethanol (TDE) (Sigma, 166782–500G) (*Staudt et al., 2007*) to exchange the water in the sample. To achieve this, retinas were incubated in different concentration of TDE buffer (10% - > 25% - > 50% - > 97%) for at least 30 min each. Then the retinas were embedded in ProLong Gold Antifade Mountant (Thermo, P36934) and gently covered with a #0 coverslip (MARIENFEL, 0100032, No.0, 18*18 mm). To avoid squeezing the retinas, we put four strips of Parafilm (Parafilm, PM999) around the retina before adding the coverslip. Some of the retinas were mounted in 97% TDE with DABCO (Sigma, 290734) after immersion into TDE. Some retinas were mounted with ProLong Gold Antifade Mountant directly after washing. Afterwards, nail polish was used to prevent evaporation and the samples were stored in darkness at 4°C.

## Retina immunohistochemistry (SMI32, CART and FOXP2)

Similar procedures were used to stain the retinas for neurofilament or CART. After fixation, freeze-cracking and blocking, primary antibody solution was added and the retinas were incubated for 5–7 days with gentle shaking at room temperature. Primary antibodies used were chicken anti-GFP (Invitrogen, A-10262, 1:500), goat anti-ChAT (Chemicon, Ab144P, 1:200), mouse SMI32 (Biolend, 801701,1:1000) and rabbit anti-CART (Phoenix, H-003–62,1:500). They were prepared in 3% NDS, 1% BSA, 0.5% TritonX-100, 0.02% NaN$_3$ in 1x PBS. Retinas were washed three times, 15 min each, in 1x PBS with 0.5% TritonX-100 before being transferred into the secondary antibody solution consisting of Alexa488 donkey anti-chicken (ImmunoJackson, 703-545-155, 1:500) and Alexa633 donkey anti-goat (Invitrogen A-21082, 1:500), Cy3 donkey anti-mouse (ImmunoJackson, 715-165-151, 1:400) and DyLight 405 donkey anti-rabbit (ImmunoJackson, 715-475-150, 1:200) with 3% NDS, 1% BSA, 0.5% TritonX-100, 0.02% NaN$_3$ in 1x PBS. Retinas were incubated in secondary antibody solution overnight at 4°C. Slices were washed three times for 10–15 min each in 1x PBS with 0.5% TritonX-100 and 1 time in 1x PBS. Mounting procedures are the same as listed above.

To stain the retina for FOXP2, we used a slightly different staining procedure. After fixation and freeze-cracking, retinas were washed three times for 10 min each in 1x PBS, followed by incubation with blocking buffer (5% NDS, 0.3% TritonX-100 in 1x PBS) overnight at 4°C. Primary antibody

against FOXP2 (abcam1307, 1:2000) was added after blocking and retinas were incubated for 5–7 days under constant gentle shaking at 4℃. They were prepared in 5% NDS, 0.3% TritonX-100 in 1x PBS. After incubation, retinas were washed three times for 15 min in 1x PBS with 0.3% TritonX-100 before being transferred into the secondary antibody solution (Alexa555 donkey anti-goat abcam150130, 1:300); prepared in 1xPBS overnight at 4℃. The second day, retinas were washed three times in 1x PBS and incubated in the second primary antibody solution for 5–7 days under constant gentle shaking at room temperature. The second primary antibodies were rabbit anti-GFP (Invitrogen, A-11122, 1:500) and goat anti-ChAT (Chemicon, Ab144P, 1:200), which were prepared in 3% NDS, 1% BSA, 0.5% TritonX-100, 0.02% NaN$_3$ in 1x PBS. After incubation, retinas were washed three times for 10 min in 1x PBS with 0.5% TritonX-100 before being transferred into the secondary antibody solution (Alexa488 donkey anti-rabbit (Invitrogen, A21206, 1:500) and Alexa633 donkey anti-goat (Invitrogen A-21082, 1:500); prepared in 3% NDS, 1% BSA, 0.5% TritonX-100, 0.02% NaN$_3$ in 1x PBS. Retinas were then washed three times in 1x PBS with 0.5% TritonX-100 and once in 1x PBS. Mounting procedures are the same as listed above.

## Brain immunohistochemistry

After removing the eyes, mice were immediately perfused with 1x PBS and 4% paraformaldehyde (PFA) and brains were post-fixed in 4% PFA overnight at 4℃. Vibratome sections (100–200 µm) were collected in 1x PBS and were incubated in blocking buffer (1x PBS, 0.3% Triton X-100, 10% Donkey serum) at room temperature for 1 hr. Then slices were incubated with primary antibodies in blocking buffer overnight at 4℃. The next day, slices were washed three times for 10 min each in 1x PBS with 0.3% TritonX-100 and incubated in secondary antibody solution diluted in blocking buffer for 2 hr at room temperature or overnight at 4℃. Primary antibodies used were rabbit anti-GFP (Thermo Fisher, A-11122, 1:500) and chicken anti-mCherry (Novus, NBP2-25158, 1:1000) and secondary antibodies used were Alexa488 donkey anti-rabbit (Thermo Fisher, A21206, 1:500–1000) and Cy3 donkey anti-chicken (ImmunoJackson, 703-166-155, 1:800–1000). Nuclei were stained with DAPI (Roche, 10236276001, 1:500) together with the secondary antibody solution. Sections were then again washed three times for 10 min in 1x PBS with 0.3% TritonX-100 and once in 1x PBS, covered with mounting medium (Dako, C0563) and a glass coverslip. For the Pbg experiments, we applied Nissl stain instead of the DAPI stain, where the Pbg can be identified as a cell-dense area. Nissl stain was applied after the secondary antibody staining. After washing with 1x PBS, the brain slices were incubated with Nissl in 1x PBS (NeuronTrace 435/455, Thermo, N21479, 1:150) for at least 20 min at room temperature. Afterwards, the sections were rinsed for 10 min in 1x PBS with 0.1% TritonX-100, followed by another two times washing for 5 min each in 1x PBS. Finally, the sections were washed on a shaker for 2 hr at room temperature or overnight at 4℃ in 1x PBS.

## Confocal microscopy

Confocal microscopy was performed on a Zeiss LSM 710 microscope. Overview images of the retina and brain were obtained with a 10x (plan-APOCHROMAT 0.45 NA, Zeiss) objective. The following settings were used: zoom 0.7, 4 × 4 tiles with 0% to 15% overlap, 2.37 µm/pixel resolution. For single retina ganglion cell scanning, we used a 63x (plan-APOCHROMAT 1.4 NA, Zeiss) objective. The following settings were used: zoom 0.7, 2 × 2 tiles or more (depending on size and number of cells) with 0% to 15% overlap. This resulted in an XY-resolution of 0.38 µm/pixel and a Z-resolution between 0.25 and 0.35 µm/pixel. The Z-stacks covered approximately 50 µm in depth.

## In vivo electrophysiology

### Surgical procedure

Eight *PV-Cre* mice of either sex at the age of 2–2.5 months were quickly anesthetized with Isoflurane (Iso-vet 1000 mg/ml) and then either maintained under Isoflurane anesthesia or injected with a mixture of Ketamine and Medetomidine (0.75 mL Ketamine (100 mg/mL) + 1 mL Medetomidine (1 mg/mL) + 8.2 mL Saline). Lidocaine (0.5%, 0.007 mg/g body weight) was injected under the skin above the skull, the animal's head was shaved, the skin and muscle tissue removed, and a titanium head plate fixed to the skull using dental cement (Metabond, Crown and Bridge). After recovery from anesthesia animals were single-housed and were administrated Buprenorphine and Cefazolin for 60 hr post-surgery (Buprenorphine 0.2 mg/kg I.P. and Cefazolin 15 mg/kg I.P. in 12 hr intervals) and

Dexamethasone (max. 0.2 ml of 0.1 mg/ml/day) depending on the condition of the animal. After this recovery phase animals were habituated for 3–4 days to the recording setup in sessions of increasing head-fixed time. One day before the first recording, the animals were anesthetized with Isoflurane and small craniotomies were performed (approximately 100 μm diameter, elongated to up to 300 μm laterally for parabigeminal coordinates and posteriorly for pulvinar coordinates). Coordinates were adjusted to each mouse's skull size based on standard coordinates for a bregma-lambda distance of 4.7 mm. Standard coordinates pulvinar: bregma −2.0/1.7 lateral. Parabigeminal nucleus: bregma −4.2/2.0 lateral.

## Data acquisition

Silicone Neuropixels probes phase 3A (Imec, Belgium) (*Jun et al., 2017*) were used to record light responses in the pulvinar and parabigeminal nucleus. The Neuropixels probes consist of a single shaft with 960 recording electrodes arranged in 480 rows with two electrodes each. The spacing between electrodes within a row (x) is 16 μm, and rows are 20 μm apart from each other (y) resulting in recording site length of 9600 μm. The 384 electrodes at the tip of the probe were recorded simultaneously in all experiments. Signals were split online into high-frequency (>300 Hz) and low-frequency (<300 Hz) and recorded separately at 30 kHz using the Neuropixels headstage (Imec), base-station (Imec) and a Kintex-7 KC705 FPGA (Xilinx). SpikeGLX was used to select recording electrodes, to calculate gain corrections and to observe and save the data. Stimulus timing information was recorded simultaneously using the digital ports of the base-station.

## Presentation of visual stimuli

A calibrated 32-inch LCD monitor (Samsung S32E590C, 1920 × 1080 pixel resolution, 60 Hz refresh rate, average luminance of 2.6 cd/m2) was positioned 35 cm in front of the right eye, so that the screen was covering 90° of azimuth and 70° of altitude of the right visual field. Visual stimuli were presented on a gray background (50% luminance), controlled by Octave (GNU Octave) and Psychtoolbox (*Kleiner et al., 2007*). The following visual stimuli were used:

### Large moving square

A black square of 53° side length moved with a speed of 150 °/sec across the screen in eight direction (0°, 45°, 90°, 135°, 180°, 225°, 270°, 315°). Each direction was repeated 10 times.

### Fast-small dot

A black dot of 4° diameter moved with 150°/s in two direction (left-right, right-left) at three different positions (center, upper quarter, lower quarter) across the screen. Each position and direction was repeated 10 times.

### Small-slow dot

Similar to the fast-small objects, a black dot of 4° diameter moved with 21°/s in two directions at three positions across the screen.

### Expansion

A small disc linearly expanded from 2° to 50° of diameter within 300 ms at the centre of the screen. The stimulus was repeated 10 times.

### Full-field 'chirp' modulation

A full-field stimulus based on the 'chirp' stimulus (*Baden et al., 2016*) starting with slow transitions gray-black-gray-white-gray (3 s at each level), followed by a temporal modulation between black and white starting at 0.5 Hz and increasing to 8 Hz over a time of 6 s. After 3 s at a gray screen, the contrast was modulated from 0% to 100% over a time period of 5.5 s at 2 Hz. The stimulus was repeated 10 times.

## Experimental design

Head-posted animals were fixed on a treadmill in front of the screen. For all pulvinar and some parabigeminal recordings, we coated the Neuropixels probe with a fluorescent dye (DiI, DiD or DiO, Thermo Fisher). The coordinates for the pulvinar (N = 4 recordings) or parabigeminal nucleus (N = 5)

were measured again and the probe was slowly lowered into the brain using a micromanipulator. Some artificial cerebrospinal fluid (150 mM NaCl, 5 mM K, 10 mM D-glucose, 2 mM NaH$_2$PO$_4$, 2.5 mM CaCl$_2$, 1 mM MgCl$_2$, 10 mM HEPES adjusted to pH 7.4 with NaOH) was used to cover the skull. Then, the probe was lowered to the desired depth. In most cases, the probe was inserted further than the targeted brain area to ensure that the whole nucleus was covered. After 20–30 min, visual stimulation and recording of neural activity was started. The setup was covered with black curtains during the whole experiment.

## Brain histology for probe location

To facilitate the identification of the pulvinar and the correct location of the probe, we injected Cholera Toxin Subunit B conjugated with Alexa488 (Thermo Fisher) into the contralateral eye to label retinal targets such as the laterogeniculate nucleus of the thalamus. Then, the brain was fixed and Vibratome sections (coronal at 100 μm) were collected in 1x PBS. The slices were washed in 1x PBS with 0.3% TritonX-100, then washed in 1x PBS and incubated for 20 min at RT with fluorescent Nissl Stain (NeuroTrace 435/455, Thermo Fisher, 1:150). Afterwards, the slices were washed in 1x PBS with 0.3% TritonX-100 and for at least 2 hr in 1x PBS. Brain slices were covered with mounting medium (Dako) and a glass coverslip, and imaged using a confocal microscope.

Probe trajectories were mapped by following DiI tracks that were typically visible across multiple slices. Recording locations along the track were manually identified by comparing structural aspects of the histological slice with features in the Allen Brain Atlas. This identification was aided by reconstruction of the track in the Allen CCF coordinates (*Shamash et al., 2018*). To achieve this, an initial guess was made of the 3D Allen CCF coordinate for each DiI track. This was aided by a control-point registration of the histological slice to an atlas slice. Once the coordinates were identified for each DiI mark along the track, a line was fitted to these coordinates in and the atlas labels were extracted from along this line. This resulted in identification of the list of brain regions each probe track and recording site passed through.

## Retinal electrophysiology

### Preparation of retinas

For in vitro recordings of retinal ganglion cells, we used mice that had been injected with herpes-simplex virus into the Pbg or pulvinar and rabies virus into the superior colliculus to label circuit specific retinal ganglion cells as described above. For pulvinar experiments, we analyzed 64 cells from 20 *Ntsr-Cre* mice. For Pbg-specific ganglion cells, we recorded 50 cells in retinas from *PV-Cre* (N = 14) or *Gad2-Cre* (N = 3) mice. Retinas were isolated from mice that were dark-adapted for a minimum of 30 min. Retina isolation was done under deep red illumination in Ringer's medium (110 mM NaCl, 2.5 mM KCl, 1 mM CaCl$_2$, 1.6 mM MgCl$_2$, 10 mM D-glucose, 22 mM NaHCO$_3$, bubbled with 5% CO$_2$/95% O$_2$, pH 7.4). The retinas were then mounted ganglion cell-side up on filter paper (Millipore, HAWP01300) that had a 3.5 mm wide rectangular aperture in the center, and superfused with Ringer's medium at 32–36°C in the microscope chamber for the duration of the experiment.

### Electrophysiology

Electrophysiological recordings were made using an Axon Multiclamp 700B amplifier (Molecular Devices) and borosilicate glass electrodes (BF100-50-10, Sutter Instrument). Signals were digitized at 20 kHz (National Instruments) and acquired using WaverSurfer software (version: 0.918) written in MATLAB. The spiking responses were recorded using the patch clamp technique in loose cell-attached mode with electrodes pulled to 3–5 MΩ resistance and filled with Ringer's medium. To visualize the pipette, Alexa 555 was added to the Ringer's medium.

### Targeted recordings using two-photon microscopy

Fluorescent cells were targeted for recording using a two-photon microscope (Scientifica) equipped with a Mai Tai HP two-photon laser (Spectra Physics) integrated into the electrophysiological setup. To facilitate targeting, two-photon fluorescent images were overlaid with the IR image acquired through a CCD camera. Infrared light was produced using the light from an LED. For some cells, z-stacks were acquired using ScanImage (Vidrio Technologies).

## Presentation of visual stimuli

Stimuli were generated with an LCD projector (Samsung, SP F10M) at a refresh rate of 60 Hz, controlled with custom software written in Octave based on Psychtoolbox. The projector produced a light spectrum that ranged from ~ 430 nm to ~ 670 nm. The power produced by the projector was 240 mW/cm$^2$ at the retina. Neutral density filters were used to control the stimulus intensity in logarithmic steps. Recordings were performed with filters decreasing the stimulus intensity by 1–2 log units. The following visual stimuli were used for retinal recordings:

### Full-field 'chirp' modulation

A full-field stimulus based on the 'chirp' stimulus (*Baden et al., 2016*) starting with slow transitions gray-black-gray-white-gray (3 s at each level), followed by a temporal modulation between black and white starting at 0.5 Hz and increasing to 8 Hz over a time of 6 s. After 3 s at a gray screen, the contrast was modulated from 0% to 100% over a time period of 5.5 s at 2 Hz. The stimulus was repeated 10 times.

### Spot-size

A black or white spot of 6 sizes (4°, 8°, 12°, 16°, 20°, 40°) was shown for 2 s at the center of the gray screen. Both the colors and the sizes were shown in random sequence.

### Large moving bar

A black bar with a width of 40° moved with a speed of 150°/sec across the screen in eight directions (0°, 45°, 90°, 135°, 180°, 225°, 270°, 315°). Each direction was repeated 5 times. The directions were randomized.

### Expansion

A black disc linearly expanded from 2° to 50° of diameter within 300 ms (150°/sec) at the center of the screen. The stimulus was repeated 10 times.

### Dimming

A disc of 50° diameter linearly dimmed from background gray to black within 300 ms (150°/sec) at the center of the screen. The stimulus was repeated 10 times.

### Looming objects

A small disc non-linearly expanded from 2° to 50° of diameter at a slow (18.5°/sec), medium (92°/sec) and fast speed (150°/sec). Each condition was repeated 10 times.

### Slow-small objects

A black disc of 4° diameter moved with 21°/sec in two direction (left-right, right-left) at the center line across the screen. Each direction was repeated 5 times.

## Morphology of patched cells

After patching, retinas were fixed and stained as described above. If the rabies labelling density allowed it, the morphology of the patched cells was imaged using a confocal microscope.

## Morphology of individual ganglion cells

To label the dendritic trees of the imaged cells in the confocal Z-stacks, we either applied a thresholding approach to identify pixels belonging to the cells, or we sent the data set to Ariadne-service GmbH (Switzerland; ariadne.ai) for tracing of the dendritic tree. The position of the ChAT-planes was extracted and used to warp both the ChAT-signal as well as the binary Z-stack of the labeled cell. Then, dendrites from other cells, noise, and axons were removed and the position of the cell body was measured. The resulting warped dendritic tree was used for further analysis such as computation of the dendritic profile, area measurements and dendritic statistics. All code can be found on github (https://github.com/farrowlab/Reinhard_2019; copy archived at https://github.com/elifesciences-publications/Reinhard_2019; https://github.com/farrowlab/ChATbandsDetection; copy archived at https://github.com/elifesciences-publications/chATbandsDetection).

### Down-sampling and binarization/tracing

The confocal Z-stacks of individual ganglion cells were denoised using the CANDLE package for MATLAB (*Coupé et al., 2012*) and down-sampled to have a resolution of XYZ = 0.5×0.5 x (0.25 to 0.35) µm per pixel and saved as MATLAB files. We then manually selected a threshold to transform the GFP-signal (i.e. the labeled cell) into a binary version where the whole dendritic tree was visible but noise was reduced as much as possible using an adapted version of the method described in *Sümbül et al. (2014a)* and *Sümbül et al. (2014b)*. Alternatively, the confocal Z-stacks were sent to Ariadne-service GmbH where the dendritic tree of each neuron was traced.

### Extraction of ChAT-positions

ChAT-band positions were either extracted manually or automatically using a convolutional neural network. For manual extraction, the ChAT-signal was smoothed using a two-dimensional standard-deviation filtering approach in the XY plane with a size of 21 × 21 pixels. The resulting Z-stacks were loaded into Fiji (*Schindelin et al., 2012*). ChAT-band positions were marked as described in *Sümbül et al. (2014a)*. Briefly, we labeled points in the ON- and OFF-band with an approximate spacing of 20 µm in X- and Y-direction. For automated labeling, an end-to-end 3D Convolutional Neural Network called V-Net with a Dice Loss Layer (*Milletari et al., 2016*) was trained on noisy greyscale images of ChAT-images, to denoise and remove any cell bodies, creating a probability map of background and foreground, with foreground being voxels that might belong to the ChAT-bands. Two smoothness-regularized-least squares surfaces were fitted to manually labeled data to train the algorithm and to create ground truth binary masks. Then, Otsu's thresholding method combined with connected component analysis was performed on the resulting probability map to automatically locate the points that belong to the ChAT-bands in new data-sets. Finally, two surfaces were independently fit to the corresponding data points to approximate the two ChAT-bands (https://github.com/farrowlab/ChATbandsDetection; copy archived at https://github.com/elifesciences-publications/chATbandsDetection).

### Warping

An adapted version of the code developed in the lab of Sebastian Seung was used to warp the GFP-signal (*Sümbül et al., 2014a*). Briefly, the ChAT-band locations were used to create a surface map, which then was straightened in 3D-space. Then, the binarized GFP-signal was warped accordingly.

### Soma position and removal of noise

After warping, the soma position was determined by filtering the GFP-signal with a circular kernel (adapted from *Sümbül et al., 2014a*). If this method detected the soma, it was used to remove the soma from the GFP-data and the center of mass was taken as the soma position. If this automated method failed, the soma position was marked manually. Afterwards, dendrites of other cells, axons, and noise were removed manually: The warped GFP-signal was plotted in side-view and en-face view in MATLAB and pixels belonging to the cell were selected manually.

### Computation of the dendritic profile and area

The distribution of the cell's dendritic tree was computed (*Sümbül et al., 2014a*). Briefly, the Z-positions of all GFP-positive pixels were normalized to be between $-0.5$ and $0.5$. Then the Fourier transform of an interpolating low-pass filter was used to filter the Z-positions. This resulted in a vector containing the distribution of pixels in the Z-direction. If necessary, this profile was used to manually remove remaining axonal or somal pixels. In this case, the dendritic profile was computed again after cleaning of the data. The area of the dendritic tree was approximated by computing a convex hull (regionprops function in MATLAB). When diameters are given, they were calculated as D = 2*(area / $\pi)^{1/2}$.

### Computation of the dendritic statistics

To compute the dendritic statistics a minimal spanning tree model was created of each imaged dendritic tree using the TREES toolbox with a branching factor of 0.4. From this tree we calculated a set of five statistics including: the mean ratio of path length and Euclidean distance; maximum metric

path length; mean branch lengths; mean path length and z-range against width of spanning field (*Cuntz et al., 2011*).

## Down-sampling of dendritic tree for plotting

For en-face plots of the dendritic arbor, they were down-sampled by calculating the local neighborhood median of all labeled pixels in patches of 50 × 50 pixels and with a sliding window of 10 pixels.

## Ganglion cell type assignment

### Preparation of dendritic profile templates

Templates of each cell type were created from the 381 traced retinal ganglion cells from EM sections of the museum.eyewire.org data basw. The EM data set was complemented by three additional data sets: Dendritic trees of examples of HD1, HD2 (*Jacoby and Schwartz, 2017*) and ventral OFF OS cells (*Nath and Schwartz, 2017*) were obtained from Greg Schwartz (Feinberg School of Medicine, Northwestern); examples of M2 (Cdh3) and sustained OFF-alpha cells (W7b) were obtained from Uygar Sümbül (*Sümbül et al., 2014a*), and dendritic profiles four FOXP2-positive cell types were extracted from *Rousso et al. (2016)*. Except for the FOXP2-positive cells, all data were processed in the same way as our data (warping, removal of axons) and average dendritic profiles were calculated for each cell type resulting in 56 profile templates.

### Correlation measurements

For each of our traced retinal ganglion cell, we calculated the linear correlation coefficient (corrcoef function in Matlab) and Euclidean distance (pdist function in Matlab) of its dendritic profile to each of the 56 templates. If the molecular identity of the cell was known, the set of compared templates was reduced to matching candidates, for example to the alpha cells of the EM data set (cluster 1wt, 4ow, 6sw, 8w) and the W7b profile of the Sümbül data set for an SMI32$^+$ cell. All clusters with a correlation coefficient or an Euclidean distance above a set threshold were considered as potential types for this cell. Thresholds were defined as the squared lower quartile of coefficients/distances of all molecularly identified cell types. For molecularly identified cells without any correlation or distance above threshold, the best match was kept. Other cells without any correlation coefficient or distance measurement above the threshold were assigned as non-classifiable.

### Decision tree

The correlation and distance measurements strongly reduced the number of potential matching templates. Cells were then assigned to one of the remaining templates based on a decision tree considering aspects of the dendritic profile, dendritic and soma size, and principal component analysis of the complete dendritic tree (pca function in Matlab). All cluster numbers (C-) used in the following text refer to the EM clusters in *Bae et al. (2018)*. Below is a detailed description of how cells lying within the same stratification were assigned.

*Cells stratifying below the ON-ChAT band:* a) C-85 separates from others as it has an additional peak in its dendritic profile that lies between the ChAT-bands. b) C-9w separate by their very large dendritic tree. c) C-8n, C-8w, C-9n differ from the remaining candidates as they lack dendrites above the OFF-ChAT-band. Within these three clusters, the alpha cells forming C-8w are distinguished by their very large soma. C-8n and C-9n could not be further distinguished. d) Of the remaining potential candidates, C-82wi and C-82wo are significantly larger than the other types. e) C-72 and C-73 were distinguished from each other based on the principal component analysis of their complete dendritic tree.

*Cells stratifying between the ChAT bands:* a) Within clusters with dendrites close to the ON-ChAT-band, only C-6t (Fmidi-ON cells) extend their dendrites below the ChAT-band (see also *Rousso et al., 2016*); to distinguish C-6sn and C-6sw (ON transient alpha cells), C-6sw have considerably bigger dendritic trees. b) Cells in C-5to and C-63 have particularly broad dendritic profiles, which are distinguished from each other as the dendrites of C-63 extend below the ON-ChAT-band and above the OFF-ChAT-band, whereas the dendrites of C-5to do not. c) Of the clusters with dendrites around the center of the ChAT-bands, C-5si is biased towards the ON-ChAT-band. Cells that fit best to either of the two HD cell types obtained from the Schwartz lab were assigned to cluster

C-5si. d) The few cells with dendrites extending above the OFF-ChAT-band, potential C-5ti or C-51, could not be clearly distinguished from each other. e) Of cells with dendrites closer to the OFF-ChAT-band, the trees of C-4ow (OFF transient alpha cells) are substantially bigger than of the other cell types, C-4i and C-4on, which could not be distinguished from each other.

*Cells stratifying above the OFF-ChAT band:* a) Cells in C-25 separate from other cells in this group based on an additional profile peak between the ChAT-bands. b) C-27 and C-28 contain the only cells with dendrites extending below the ON-ChAT-band. They were distinguished from each other based the principal component analysis of their complete dendritic tree. c) Of the clusters with dendrites extending below the OFF-ChAT-band, C-1wt (OFF sustained alpha cells) are considerably larger than the other cells and C-2an contains substantially smaller cells than C-3i and C-3o. d) Of the cells with dendrites far above the OFF-ChAT-band, C-1ws (M1 cells) are the largest cell type in the retina. C-1ni and C-1no were not distinguishable.

Cells that did not fulfil the criterions of any cluster within their group were considered 'non-classifiable'. We did not distinguish between different types of ON-OFF-DS (37 c,d,r,v) and ON-DS cells (7id,ir,iv,o).

## Visual inspection

After assigning each cell to an EM-cluster, the en-face and side-views of all cells in a given cluster were visually inspected. For potential outliers, we compared the dendritic statistics of this cells to the statistics of its current and two next best candidate cell types. If the statistics were closer to the average statistics of an alternative cluster, this cell was reassigned. In addition, some cells of C-8n and C-8w could not be clearly assigned based on their soma. However, the sustained alpha cells in C-8n have a particular soma shape and pattern of dendritic roots. In these cases, the detailed dendritic and soma morphology in the original confocal scans were inspected and reassigned by an expert.

## Size distribution analysis

For retinotopic size distribution calculations, we computed a moving median diameter within a circular window of 250 µm radius, using a step size of 100 µm. The resulting $50 \times 50$ median size matrix was convolved with a gaussian with sigma = 200 µm (using MATLAB function fspecial and nanconv).

## Quantification of SMI32$^+$ cells, CART$^+$ cells and FOXP2$^+$ cells

### Numbers of double-labeled cells

To quantify the number of double-positive cells for CART/GCaMP6s and SMI32/GCaMP6s, we scanned a z-stack (1 to 5 µm Z-resolution) of the whole retina using the confocal microscope with an 10x objective. Images of the anti-CART, SMI32 or FOX2 and the anti-GFP staining were opened in Fiji. For counting CART$^+$ cells, cells were marked using the point tool and counted manually. Note that the anti-CART antibody also labels a group of amacrine cells, therefore the complete Z-stack should be checked for each CART$^+$ cell to make sure that the labelling truly overlaps with the anti-GFP signal. The CART expression pattern was consistent with previous reports (*Kay et al., 2011*). In total we counted three retinas for parabigeminal experiments and six retinas for pulvinar experiments. For SMI32 stainings, cells were counted manually using the cell counter plugin. In total, we counted three retinas for parabigeminal experiments and four retinas for pulvinar experiments. For FOXP2 stainings, cells were counted manually using the cell counter plugin. In total, we counted five retinas for parabigeminal experiments and eight retinas for pulvinar experiments.

### Numbers of cells for types of alpha cells

To test which of the four alpha cell types were part of each circuit, we acquired small high-resolution Z-stacks (2.5 µm/pixel) of XY = 103×103 µm size (128 × 128 pixel, 63x objective) covering the full depth of the dendritic tree and centered around the soma of 91 SMI32$^+$ / GCaMP6s$^+$ cells in n = 3 retinas from parabigeminal experiments and 90 SMI32$^+$ / GCaMP6s$^+$ cells in n = 3 retinas from pulvinar experiments. We plotted top and side views of each Z-stack in MATLAB and manually decided for each cell if it was a sustained ON-alpha cell (dendrites below the ON- ChAT band), a transient ON-alpha (dendrites just above the ON- ChAT band), a transient OFF-alpha (dendrites just below or on the OFF- ChAT band) or a sustained OFF-alpha cell (dendrites above the OFF- ChAT band).

## Spike sorting

The high-pass filtered in-vivo data was automatically sorted into individual units using SpyKING CIR-CUS (*Yger et al., 2018*). The following parameters were used: cc_merge = 0.95 (merging if cross-correlation similarity > 0.95), spike_thresh = 6.5 (threshold for spike detection), cut_off = 500 (cut-off frequency for the butterworth filter in Hz). Automated clustering was followed by manual inspection, merging of units if necessary and discarding of noise and multi-units using phy (https://phy-contrib.readthedocs.io). Units were evaluated based on the average waveform shape and auto-correlogram. Only cells with <1% of inter-spike intervals of ≤1 ms were considered. In addition, we tested if their cross correlograms with nearby neurons showed evidence for being spikes from the same neurons (*Segev et al., 2004*; *Yger et al., 2018*).

## Analysis of in vivo recordings

Unless otherwise noted, firing rates were calculated as the number of spikes in 50 ms bins averaged across the 10 stimulus repetitions. Z-scores were calculated as the number of standard deviations from the mean spontaneous activity before stimulus onset. All sorted units were grouped into cells with a maximal response amplitude > 2 standard deviations above the mean spontaneous firing rate ('potentially responding') and cells without such a peak ('non-responding'). The activity to each stimulus repetitions was inspected for the 'potentially responding' cells to identify truly responding cells manually, which then were used for further analysis, average response calculations and visualization. For small stimuli shown at three different locations and moving in two different directions, only the strongest response was considered for population analysis.

### DSI

Direction-selectivity was calculated based on the summed, back-ground subtracted activity during the time from the onset of the fast moving square until the end of the presentation for each direction α. These eight response measurements $R_k$ were normalized to the maximum and the DSI was calculated according to: $\sum_k R_k \cdot e^{a_{ik}} / \sum_k R_k$.

### Half-width of response to small, slow dot

Mean firing rates for each cell were background subtracted and the MATLAB function findpeaks was used to find the half-width of the highest peak.

## Analysis of patch-clamp recordings

The loose-patch extracellular recording traces were high-pass filtered. Events that exceeded an amplitude threshold were extracted. Unless otherwise noted, firing rates were calculated as the number of spikes in 50 ms bins averaged across the 5–10 stimulus repetitions.

### Chirp

Average responses were calculated based on the mean number of spikes during the stimulus across 10 trials.

### Frequency responses

Spikes produced in response to the frequency part of the chirp stimulus were binned in 1 ms bins and the Fourier Transform was calculated using the Matlab function fft. The mean Fourier Transform amplitude for different frequency ranges was calculated for *Figure 6A*.

### Spot-size tuning curve

Firing rates were background subtracted and peak responses during the first 0.4 s after each stimulus onset were calculated and used to plot a spot-size tuning curve.

### DSI/OSI

Direction-selectivity was calculated as for the in-vivo recordings. Firing rates were background subtracted and peak responses during the first 1 s after each stimulus onset were calculated. The direction-selectivity of a ganglion cell was defined as the vector sum of these peak responses for each of the eight different directions α. These eight response measurements $R_k$ were normalized to the

maximum and the DSI was calculated according to: $\sum_k R_k \cdot e^{a_{ik}} / \sum_k R_k$. Similarly, the orientation-selectivity index (OSI) was calculated according to: $\sum_k R_k \cdot e^{a_{2ik}} / \sum_k R_k$.

## Assigning recorded retinal ganglion cells

To assign patched ganglion cells (n = 123) to the anatomical clusters, we assigned their 'chirp' responses to templates of identified 'chirp' responses. The set of templates included all 49 clusters of *Baden et al. (2016)* and the average response of patched cells that were assigned to one of our 14 clusters based on their morphology. We had such cells with both anatomy and physiology for clusters 1, 3, 4, 5, 6, 7, 8, 11, 12, 13, 14 (n = 48 cells in total). For comparison with the calcium imaging traces of the published data set, we convolved the chirp responses of the remaining 75 patched cells with the Kernel of the calcium indicator ogb1 used in *Baden et al. (2016)*. Three distance measurements were calculated for each of these patched cells to compare them to the 49 + 12 templates. The distance measurements consisted of linear correlation coefficient (corrcoef function in Matlab), Euclidean distance (pdist function in Matlab) and residuals (subtraction of the response from each template). If a cluster was in the top two for at least two distance measurements, the cell was assigned to this cluster. If the distance measurements did not agree with each other, the best cluster with the highest linear correlation coefficient was taken. For each of our anatomical clusters, we plotted and further analyzed the visual response if it contained at least four patched cells. This led to the analysis of a total of 93 patched cells.

## Comparison of in vitro and in vivo data

To compare the response properties of different retinal ganglion cell types and neurons in the Pbg and pulvinar, we calculated z-scores for each responding neuron as described above. Median firing rates were plotted for the different brain nuclei and retinal ganglion cell types.

## Cell body size measurements

To separate sustained ON-alpha cells from non-alpha cells, we loaded the original z-stack into Fiji, calculated a maximal projection and used the ellipse tool to fit an ellipse to the cell body and measure its area.

## Statistics

To compare dendritic tree diameter distributions, we applied the Kolmogorov-Smirnov test (kstest2 function in MATLAB). Medians were compared by the Wilcoxon rank sum test (ranksum function in MATLAB). We used Pearson correlation (corr function in MATLAB) to test for significant gradients in the retinotopic distribution of dendritic tree diameters.

Two tests were used to assess the statistical significance of different sampling of each cell type or response feature by the two circuits. Either, a two-proportion z-test was used, where for each cell type we computed the test parameters, k = 'number of cells assigned to a cell type for each pathway,' and p = 'proportion of cell type in total population.''. We then performed a two-tailed z-test to determine if the proportion of cells of a particular cell type deviates from our null hypothesis that the proportion in the total population should be the same as in each pathway. After p value correction for false discovery rate (*Benjamini and Yekutieli, 2001*), p values were considered significant at alpha = 0.05. Or, we performed a bootstrap analysis of whether the percent difference (%LP - %Pbg / %LP + %Pbg) of sampling of each ganglion cell by the two circuits is different from zero. To accomplish this, distributions were estimated using 10,000 repetition of random sampling from all our retinas with replacement. After p value correction for multiple comparisons (*Benjamini and Yekutieli, 2001*), p values were considered significant at alpha = 0.05.

## Acknowledgements

We thank Keisuke Yonehara (DANDRITE, Aarhus, Denmark) for supplying the Ntsr1-GN209Cre mice, Martón Balogh for performing patch-clamp experiments; Norma Kühn and João Couto for reading the manuscript, as well as João Couto and Cagatay Aydin with help setting up Neuropixels hardware. Grants are as follows: Marie-Curie CIG (631909) and FWO Research Project (G094616N) to KF.

This project has received funding from the European Union's Horizon 2020 research and innovation programme under the Marie Skłodowska-Curie grant agreement No 665501 to KR (12S7917N). CL is funded by the Chinese Scholarship Council.

## Additional information

### Funding

| Funder | Grant reference number | Author |
| --- | --- | --- |
| Fonds Wetenschappelijk Onderzoek | G094616N | Karl Farrow |
| FP7 People: Marie-Curie Actions | 631909 | Karl Farrow |
| H2020 Marie Skłodowska-Curie Actions | [PEGASUS]$^2$ Marie Skłodowska-Curie Fellowship 12S7917N | Katja Reinhard |
| China Scholarship Council | | Chen Li |

The funders had no role in study design, data collection and interpretation, or the decision to submit the work for publication.

### Author contributions

Katja Reinhard, Conceptualization, Data curation, Formal analysis, Funding acquisition, Validation, Investigation, Visualization, Methodology, Writing—original draft, Writing—review and editing; Chen Li, Conceptualization, Data curation, Funding acquisition, Investigation, Visualization, Methodology, Writing—original draft, Writing—review and editing; Quan Do, Software; Emily G Burke, Investigation; Steven Heynderickx, Methodology; Karl Farrow, Conceptualization, Software, Formal analysis, Supervision, Funding acquisition, Investigation, Visualization, Methodology, Writing—original draft, Project administration, Writing—review and editing

### Author ORCIDs

Katja Reinhard (iD) https://orcid.org/0000-0002-8719-7445
Chen Li (iD) http://orcid.org/0000-0002-5117-986X
Karl Farrow (iD) https://orcid.org/0000-0003-1409-096X

### Ethics

Animal experimentation: All animal procedures were performed in accordance with standard ethical guidelines of KU Leuven and European Communities Guidelines on the Care and Use of Laboratory Animals (004-2014/EEC, 240-2013/EEC, 252-2015/EEC).

### Decision letter and Author response

Decision letter https://doi.org/10.7554/eLife.50697.sa1
Author response https://doi.org/10.7554/eLife.50697.sa2

## Additional files

### Supplementary files

• Transparent reporting form

### Data availability

Data is available via the Open Science Framework: https://osf.io/b4qtr/. In particular, the morphology of all ganglion cells is provided including extra information of molecular labels and physiological recordings are available. This will enable the recreation of Figure 1, 2 and 5. In addition, we have made the spike times of all recorded neurons in the retina and central brain regions included in the paper available, enabling recreation of Figure 4, 5, 6 and 7. Code is also available at https://github.

The following dataset was generated:

| Author(s) | Year | Dataset title | Dataset URL | Database and Identifier |
|---|---|---|---|---|
| Farrow K | 2019 | A projection specific logic to sampling visual inputs in mouse superior colliculus | https://osf.io/b4qtr/ | Open Science Framework, b4qtr |

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
