## [Decision Letter]

[Editors’ note: a previous version of this study was rejected after peer review, but the authors submitted for reconsideration. The first decision letter after peer review is shown below.]

Thank you for submitting your work entitled "A projection specific logic to sampling visual inputs in mouse superior colliculus" for consideration by *eLife*. Your article has been reviewed by three peer reviewers, including Fred Rieke as the Reviewing Editor and Reviewer #1, and the evaluation has been overseen by a Senior Editor. The following individuals involved in review of your submission have agreed to reveal their identity: Samuel Solomon (Reviewer #3).

Our decision has been reached after consultation between the reviewers. The reviewers were all enthusiastic about the general aim of the research and the potential significance of the results. However, several critical limitations were raised in the reviews, and these are sufficiently important that we cannot consider the work further for publication in *eLife*. Should you be able to address these concerns fully, we would be happy to reconsider the manuscript. We note it would be considered as a new submission rather than a revision.

All of the reviewers read each other’s reviews and agreed with the points raised in all of the reviews. Three important points are summarized below. The individual reviews (also below) detail these and other issues.

1) The pathway specificity of the ganglion cell projections and the conclusions drawn from it needs to be described more carefully, with attention to the result that the specificity is not perfect. Also, important here is consideration of the relatively small number of cells in some cases. More detailed analysis of this data should include statistical tests of the significance of differences in projection patterns.

2) The paper needs to be more transparent in describing the anatomical and physiological clustering approaches and their relationship with previous work. Specific examples of these issues are raised in the individual reviews. Again, statistical tests will be important here.

3) The viral tracing and the Ntsr1-GN209-Cre line both have potential limitations/biases which should be considered in interpretation of those results.

Reviewer #1:

This paper examines an important and timely issue: the anatomical and functional connectivity from retina to superior colliculus to two target areas of the colliculus, the pulvinar and the parabigeminal nucleus. The paper uses a powerful combination of viral tracing, anatomy and electrophysiology to show that there is some specificity in this circuitry. The breadth of approaches is impressive, and I am quite enthusiastic about the overall aim of the work. There are several points, however, where the data does not fully support the conclusions drawn. These are detailed below.

1) Anatomical selectivity of circuits.

The selectivity of the two circuits studies is described in an all-or-none fashion when the data is not so clear. For example, cells that are split something like 70%/30% with respect to the two brain regions are discussed as having near perfect selectivity (e.g. Abstract, also see subsection “Clustering of ganglion cell anatomy reveals selective sampling by the colliculo-parabigeminal and the colliculo-pulvinar circuit”). The data itself shows a more nuanced segregation, with all cell types studied having some projections to both brain areas. The text should reflect the data much more closely. This is an issue throughout the paper.

2) Subgroups of OFF-alpha cells.

The evidence for subgroups of alpha cells, and corresponding specificity in their projections, is interesting but underdeveloped. In the case of the OFF-transient alpha cells, there is evidence for anatomical tiling based on SMI32 labeling (Bleckert, 2014). Did you record from cells in the different subgroups in the same retina? Given how much is known about these cell types from past work, and given the strong role that tiling plays in what we know about the organization of different RGC types, more evidence is needed to reach the conclusion that the cells can be subdivided into functionally distinct groups.

3) Comparison of functional properties of RGCs and downstream projections

Figure 7 compares responses to expanding spots of RGCs with downstream projection areas. Interpretation of this comparison difficult since the kinetics of the retinal responses are slower than those in the brain (particularly the Pbg). I would consider saving this comparison for another paper where it can be developed more completely.

The recent paper from Roson et al., (2019) is highly relevant for the present work and should be included in the Discussion.

Reviewer #2:

The authors studied the "logic" of retinal projections to the SC according to the collicular output pathways. They used transsynaptic labeling to trace the RGCs that innervate the SC neurons projecting to the LP or the Pbg, then characterized the morphology and physiology of the labeled RGCs. Their results suggest a "projection-specific" logic. They also performed in vivo recording of the LP and the Pbg to correlate the response properties of these structures with those of the RGCs that target them di-synaptically. Overall, the topic is important, and the experiments/analysis are largely appropriate. But the authors tend to over-state their findings from a limited dataset. A number of major and minor issues need to be addressed.

Major concerns:

1) From the RGC morphological analysis, 12 clusters were observed. According to Figure 3, ALL of them project to both structures, with different degrees of bias. But the authors over-interpret these results by stating "uniquely sampling" (Abstract). This is not supported by the data and would be extremely misleading to casual readers of the article, especially given the relatively small dataset. For example, cluster 8 in Figure 3 only contained 8 cells, and 2 of them projected to Pbg and 6 to LP, and yet it was considered one of the "clusters that are almost exclusively part of the pulvinar circuits". Similarly, the number of samples in the RGC physiology experiments are rather small for most clusters.

2) The efficacy and potential sampling bias of rabies tracing need to be addressed. This is necessary in order to conclude that the two circuits "sample from a limited set, ~14 out of more than 30".

3) The use of Ntsr1-GN209-Cre mice needs to be raised as a potential technical concern. It is possible that the Ntsr1 positive cells are not the only SC cells that project to the LP, As a result, using these cre mice could under-estimate the RGC types that target SC-pulvinar pathway.

Reviewer #3:

Overall, the study is well motivated, well constructed, and well presented. It is really positive to see the morphology of all the RGCs in supplementary material. The manuscript primarily describes viral-mediated morphology and functional imaging of retinal ganglion cells that connect via the superior colliculus to two important brain nuclei (LP, Pbg); this morphological and functional dataset is important and impressive. In addition, the authors record extracellularly from neurons in the LP and Pbg (with unknown inputs) and measure response to several stimuli.

Major points:

1) The first part seems to imply that a) there are ~12 morphological classes of RGCs projecting to the 2 pathways, a subset of the ~30 classes previously demonstrated in mouse retina, and b) that some distinct classes of RGC project to LP or Pbg. First maybe I am missing something, but it is not clear to me if the clustering applied to this morphological data would reveal 30 classes across the retina – it may reveal 14, or it may reveal 300. Without some norm for the clustering we don't know what fraction of mouse RGC classes are identified in these analyses. Second there is no way to know where these RGCs project to outside the two target areas, so we do not know if for example the Pbg receives a unique sample (as implied in the Abstract) or if the Pbg pathways substantially overlap other (unsampled) pathways. Third, it is not clear to me from the manuscript how confident the authors are in estimating overlap in LP and Pbg RGC pathways – in Figure 3 for example, clusters 4,7 and 12 are inferred to be part of the Pbg pathway but clusters 5,9 and 11 are inferred to innervate both circuits, but these all appear to have similar biases or have so few neurons that the confidence intervals on the% to each pathway must be very large. The manuscript needs much greater clarity about how these inferences are made, and the statistical support that they have.

2) I was confused by how the retinal functional measurements are clustered and categorised. The authors say they identified 12 groups (subsection “Functional classes of retina ganglion cells show refined pathway selectivity”) but do not say how these were identified. For example, it is not clear to me why Group (ii) is a sustained OFF cell when the onset response appears transient, and on what basis it is distinguished from Group (iv). Some more explanation of the response to these stimuli would be useful; claims of functional differentiation between cells, and therefore whether or not particular functional classes project to particular brain areas, would also seem to require statistical support.

3) The in vivo data though interesting is difficult to relate to the rest of the data. I do not think that it enhances this paper. e.g. (1) The authors note latency differences between Pbg and LP but there also appear to be rate differences; latency is often longer in weak responses and a fair comparison would need to match the firing rates of the two populations; (2) The authors note the presence of direction selective units in Pbg but not LP (Figure 6A) but as there are only 12 units in LP I am not sure how confident one should be in this. In addition, it is clear from other work that LP organisation depends on e.g. AP location – were the injection sites and recordings matched in location? As for the other data I could not find confidence intervals on the estimates of functional properties in Pbg and LP, making it difficult to know how well they can be distinguished.

4) The Supplementary file 1 does a good job of trying to align the classes identified by the authors with those identified by others. It is much appreciated. I think that a similar approach is necessary to be able to join the different data sets presented here – it is often unclear what the basis for deciding on the category is for a particular dimension of analysis (functional, morphological, immunostaining) and how categories are matched of categories across dimensions.

5) There are too many unsupported and/or ambiguous phrases (non exhaustive list: Abstract: "projection specific", "uniquely sampled", "correlated well", "mechanistic basis for selective triggering of visually guided"; Introduction" "we found strong specificity"; Results section: "exclusively part of"; "very strong bias"; "rather small"; "relatively large"; "exceptionally broad"; "striking selectivity"; "had a tendency towards"; Discussion section: "clear segregation", "strong preference", "could be explained by selective sampling of different retinal ganglion cell types", "dedicated set of connections", "confidently identify", "high degree of regularity" etc). I believe the data is quite clear – and the wording should reflect the data the authors show, and the appropriate statistical analyses that they apply.

[Editors’ note: what now follows is the decision letter after the authors submitted for further consideration.]

Thank you for submitting your article "A projection specific logic to sampling visual inputs in mouse superior colliculus" for consideration by *eLife*. Your article has been reviewed by three peer reviewers, including Fred Rieke as the Reviewing Editor and Reviewer #1, and the evaluation has been overseen by Joshua Gold as the Senior Editor. The following individuals involved in review of your submission have agreed to reveal their identity: Samuel Solomon (Reviewer #2).

The reviewers have discussed the reviews with one another and the Reviewing Editor has drafted this decision to help you prepare a revised submission.

One salient point emerged from the discussion among reviewers: a need to present the results about projection bias in a way more closely tied to the original data. Currently the data is subjected to a bootstrap analysis to determine if the number of projections to a given area is significantly different from 0, and the results are then summarized (e.g. in Figure 8) in an all-or-none fashion. But this does not accurately represent the fact that none of the RGC types projects exclusively to one or the other SC target area. We all agreed that the data should be presented in a more straightforward way – e.g. as numbers of projections or the ratio of those numbers (with confidence intervals). Significance tests (probably non-parametric) could be applied to whether the number of projections to the two areas differ significantly. More details about those concerns, as well as several other issues, can be found in the individual reviews below.

Reviewer #1:

This is a revised paper about the projections from retina to SC to two SC targets: the pulvinar and the parabigeminal nucleus. The paper uses an impressive array of circuit tracing and electrophysiological approaches to show that the retinal ganglion cells that provide (via the SC) input to these two target areas differ considerably. The paper has improved considerably in revision, and the central message is very clear and well supported by the data (with one important exception – see below). The authors should be congratulated on both the work and on the strength of the revisions. I have a few suggestions below for clarity.

Subsection “Some visual responses of pulvinar and parabigeminal nucleus are explained by selective innervation of retinal ganglion cell types” (and later in the Discussion section): I don't think the lack of responses to full-field stimuli requires a nonlinearity. For example, you could have a linear summation of responses with oppositely signed weights from two sets of cells with similar responses to the chirp (or other full field) stimuli. I believe that in the LGN work the ganglion cells were combined with mostly or exclusively positive signs, so I think the difference here is that you either need a nonlinearity or a combination of responses with differing signs so that you can get cancelation.

Reviewer #2:

The paper is even stronger and the authors have addressed most of my previous concerns. I think this is an important set of experiments with strong anatomical conclusions and less strong functional conclusions. The authors inferences generally reflect these, are fair and justified by the data with some small exceptions.

1) In the Abstract: "These findings suggest that projection specific sampling of retinal inputs forms a basis for the selective triggering of behaviours by the superior colliculus". I don't think that this statement is sufficiently supported by the work, as the functional distinction is not clear – for example, the authors show that despite different retinal inputs, looming stimuli apparently activate both Pbg and LP pathways (not obviously consistent with the selective triggering of behaviours), while the potent chirp stimuli for the retinal ganglion cells apparently have no counterpart centrally and probably not behaviourally. I would think this needs rewording to offer speculation not conclusion (e.g. "These findings open the possibility that projection specific sampling of retinal inputs helps form a basis for the selective triggering of behaviours by the superior colliculus"). Similarly, the first paragraph of Discussion section accentuates the difference populations that are sampled, and ignores the similarities, creating the unfortunate impression that they are non-overlapping inputs.

2) Subsection “Functional properties of retinal ganglion cells support anatomical classification” is less easily read than the other sections. I think that the authors have reconstructed 23 dendritic fields, assigned each to one of the classes, averaged the responses within each class, then assigned the non-reconstructed neurons to the same class on the basis of the similarity in their functional properties. There is some risk of circularity here, depending on the question. I am not too worried about that here, but I do think the authors need to be careful in presenting the physiological data because it may generate more certainty in anatomical-physiological correlations than is warranted. This can be circumvented by making it clear in the legends to Figure 4 and Figure 7. Also, the 'N=' values on the left of the rows in Figure 4 should be adjusted to report the number of anatomically identified units in each cluster to make sure there is no confusion – the total N could be reported in the physiology column. N values should also be reported in Figure 7.

One point- I can't quite work out the numbers here – in subsection “Assigning recorded retinal ganglion cells” the authors state that 11 clusters had at least 1 cell with anatomy and physiology, then state they measured chirp responses of 123 patched cells (should this be 23?), and then state that they further analyse only clusters with at least 4 patched cells. In Figure 4 there 7 clusters (minimum number of units = 28). I therefore assume that the authors mean that the anatomical-physiological correlation of a cluster can be based on <4 patched cells with both anatomical and physiological measurements. If not, please clarify, if so, please make that clear, and simply state the numbers of cells with combined a/p in each cluster.

3) Subsection “Retrograde transsynaptic labelling of retinal ganglion cells”: Some moderation on the Ntsr1 line is still needed. The Zhou et al., (2017) study cited shows fairly clearly only WF-like cells, probably though not certainly Ntsr1, in their Figure 2B but that is a contralateral section (ie. showing the contralateral projection to LP). Additional more superficial cells, which may or may not be WF, appear to be labelled in e.g. Figure 8C after LP injections. I've not strong opinions about whether there are or are not multiple subclasses of SC neurons (including multiple subclasses of WF neurons) but I would argue that we don't yet know with certainty.

Reviewer #3:

The authors have significantly revised the manuscript by adding more data, performing new analysis, and including necessary discussion. The manuscript is much improved as a result. My only remaining issue is the "all-or-none" statement regarding projection specificity. I disagree with the new bootstrap analysis used to support their conclusion and I am skeptical of the procedure to analyze percentage instead of the absolute numbers. My concerns are elaborated below.

Using "cluster 3" as an example, 11 cells were found, with 8 in Pbg experiments and 3 in LP experiments, out of the 196 to Pbg and 354 to LP (by the way, was there 550 or 599 total cells in the data set? The numbers in the manuscript are inconsistent).

From these numbers, the authors calculated the percentage to be ~4% to Pbg and <1% to LP. I agree that there appears to be a preference for RGCs in this cluster to project (indirectly via SC) to Pbg, and I also think such a result is worthy of a publication. But the authors went with a bootstrap analysis that shows the percentage to the LP is NOT different from 0, thus stating they "almost exclusively" project to Pbg (and complete exclusivity in the summary figure in Figure 8). This is just wrong -- cells in this cluster clearly project to LP. This statistical test is just inappropriate, and Figure 8 is dangerously misleading. Why not just illustrate the preference with lines of different thickness?

Regarding calculating the percentage, I am not sure that's appropriate or informative either. The percentage is out of the total number of labeled cells, not the likelihood of a certain type RGCs projecting to one structure over the other. Again using cluster 3 as an example, the 8 vs. 3 difference (# of labeled cells) indicate that these cells are more than twice likely project to Pbg, from the RGCs' perspective in terms of "projection logic", but not quite the 4 times difference as indicated by the percentages. Additionally, the authors acknowledge that the labeled cells in LP experiments were consistently more than in the Pbg experiments, and that this could be due to the fact that Pbg was difficult to target (Subsection “Retrograde transsynaptic labelling of retinal ganglion cells”). In other words, the percentage difference could likely be affected by experimental factors.

---

## [Author Response]

[Editors’ note: the author responses to the first round of peer review follow.]

All of the reviewers read each other’s reviews and agreed with the points raised in all of the reviews. Three important points are summarized below. The individual reviews (also below) detail these and other issues.

We appreciate the reviewers’ criticisms and suggestions. We have in this resubmission doubled the anatomical data included in the dataset. This has allowed us to address the main concerns raised by the reviewers, which focused on the lack of statistical evidence to support the arguments about pathway specificity. To address these issues, we have taken four key steps. First, we doubled the number of neurons (n = 599). Second, we created a decision tree that assigns our anatomical data to the EM reconstructions available in the EyeWire museum (http://museum.eyewire.org/; Bae et al., 2018), which provide an almost complete quantitative description of the retinal ganglion cell population. Third, we have added a set of statistical tests that independently assess the differences in proportion, and differences from 0 of how each cell type innervates the two circuits. Finally, for a subset of data we have managed to collect the anatomy and physiology for single neurons, which has allowed us to form a clear link between our anatomical, molecular and physiological datasets. For answers to specific concerns see below.

1) The pathway specificity of the ganglion cell projections and the conclusions drawn from it needs to be described more carefully, with attention to the result that the specificity is not perfect. Also, important here is consideration of the relatively small number of cells in some cases. More detailed analysis of this data should include statistical tests of the significance of differences in projection patterns.

The text has been altered to describe the degree of specificity of the ganglion cell projections more carefully. Statistical tests have been added where appropriate. Specifically, we used the z-test of two proportions to see if one pathway sampled a cell type preferentially compared to the other and a bootstrapping test to determine if a pathway sampled a statistically significant number of neurons (> 0) of a given type cell type.

2) The paper needs to be more transparent in describing the anatomical and physiological clustering approaches and their relationship with previous work. Specific examples of these issues are raised in the individual reviews. Again, statistical tests will be important here.

We have altered our approach to assigning each neuron to one of the cell-types described in the EyeWire museum based on anatomical and molecular information. This procedure is described conceptually in the Results section and in greater detail in the Materials and methods section. We believe this approach provides a transparent relationship with previous work as we are making assignments based on direct quantitative comparisons with published data.

Regarding the physiological retina data, we now assign responses to the responses of anatomically classified neurons in our own dataset in a similar approach to Roson et al., 2019 (Figure 4). We do not include physiological data where we did not have a match to an anatomically identified cell type. We believe this process provides a better link between our anatomical and physiological data and reduces the uncertainty produced by the incomplete physiological description that exist in the best anatomical datasets (e.g. Bae et al., 2018) and incomplete anatomical descriptions in the best physiological datasets (e.g. Baden et al., 2016).

3) The viral tracing and the Ntsr1-GN209-Cre line both have potential limitations/biases which should be considered in interpretation of those results.

We agree and have considered this carefully. We have in the discussion now added a section that directly addresses the potential limitations/biases of both the viral tracing techniques used and the use of Ntsr1-GN209-Cre (Discussion section).

Reviewer #1:This paper examines an important and timely issue: the anatomical and functional connectivity from retina to superior colliculus to two target areas of the colliculus, the pulvinar and the parabigeminal nucleus. The paper uses a powerful combination of viral tracing, anatomy and electrophysiology to show that there is some specificity in this circuitry. The breadth of approaches is impressive, and I am quite enthusiastic about the overall aim of the work. There are several points, however, where the data does not fully support the conclusions drawn. These are detailed below.1) Anatomical selectivity of circuits.The selectivity of the two circuits studies is described in an all-or-none fashion when the data is not so clear. For example, cells that are split something like 70%/30% with respect to the two brain regions are discussed as having near perfect selectivity (e.g. Abstract, also see subsection “Clustering of ganglion cell anatomy reveals selective sampling by the colliculo-parabigeminal and the colliculo-pulvinar circuit”). The data itself shows a more nuanced segregation, with all cell types studied having some projections to both brain areas. The text should reflect the data much more closely. This is an issue throughout the paper.

We agree that the text did not reflect the data as accurately as it should. In the new version we have attempted to make sure the text reflects the data as accurately as possible and separated interpretations and conclusions from descriptions of the results. However, we maintain that some of the data does warrant an all-or-none description, for a few of the cell types. We believe this is now clear in the increased dataset and clearer analysis (Figure 2). These statements are now supported with statistical tests to determine if each cell type has a biased distribution between the two circuits (two-proportion z-test), and whether each cell type is represented with significantly more than 0 cells in a given circuit (bootstrap test). We found that each cell-type that showed a statistically negative bias to one of the circuits was also found to not innervate that circuit with enough neurons to be statistically greater than 0. If these two tests are significant, then the cell type is considered to uniquely innervate one of the circuits and not the other (Figure 2). In the discussion we add a paragraph about the caveats to these statements (subsection “Retrograde transsynaptic labelling of retinal ganglion cells”).

2) Subgroups of OFF-alpha cells.The evidence for subgroups of alpha cells, and corresponding specificity in their projections, is interesting but underdeveloped. In the case of the OFF-transient alpha cells, there is evidence for anatomical tiling based on SMI32 labeling (Bleckert, 2014). Did you record from cells in the different subgroups in the same retina? Given how much is known about these cell types from past work, and given the strong role that tiling plays in what we know about the organization of different RGC types, more evidence is needed to reach the conclusion that the cells can be subdivided into functionally distinct groups.

We agree, we also found the different subgroups to be very interesting. We also agree that the analysis was underdeveloped and the ability to demonstrate evidence of tiling is critical. Unfortunately, we have not been able to record enough neurons in the same retina to perform the adequate analysis of the tiling. In addition, no evidence exists that relates these different OFF-alpha subtypes to different anatomical classes (Baden et al., 2016). Given our inability to adequately address this issue we have removed this separation from the paper. One point to highlight is that we no longer cluster the physiological data independently from the anatomical data. However, for the different alpha subtypes (sustained OFF, transient OFF and sustained ON) we do show that the size distributions of the different alpha cell types is consistent with biases observed by Bleckert et al., 2014 (see Figure 5), and we demonstrate that the transient responses of transient OFF-alpha cells show an inhomogeneous distribution that is consistent with that shown by Warwick et al., 2018 (see Figure 5).

3) Comparison of functional properties of RGCs and downstream projectionsFigure 7 compares responses to expanding spots of RGCs with downstream projection areas. Interpretation of this comparison difficult since the kinetics of the retinal responses are slower than those in the brain (particularly the Pbg). I would consider saving this comparison for another paper where it can be developed more completely.

We agree that the causal relationship between functional properties of RGCs and downstream projections is difficult to make. However, we still believe reporting two of the relationships is meaningful. First, strong direction-selective responses were only observed in the Pbg-circuit, which is matched by its specific innervation by ON-OFF direction-selective retinal ganglion cells. Second, strong responses to expanding dark disks is observed in both brain structures and this is matched by strong responses in cell-types that innervate both circuits. We believe these two relationships are worth reporting and we have added a paragraph to the Discussion section about the highly non-linear relationships that exist and difficulties we have in drawing comparisons.

One caveat to this interpretation is that putative F-mini (ON and OFF) neurons innervate the pulvinar. We recorded direction-selective responses in the parabigeminal nucleus but not in the pulvinar, which was mirrored by the selective innervation of ON-OFF direction-selective neurons to the colliculo-parabigeminal nucleus. However, Fmini-ON, which innervate both circuits, and Fmini-OFF cells, which selectively innervate the colliculo-pulvinar circuit, have been reported to be direction-selective (Rousso et al., 2016).

There are three reasons why we believe that this selectivity may not make a major contribution to direction-selective responses in the superior colliculus and its downstream targets. First, it has been demonstrated that direction-selective responses in the superior colliculus rely on the inhibitory output of starburst amacrine cells (Shi et al., 2017), where starburst amacrine cells are responsible for the direction-selective responses of ON-OFF but are unlikely to contribute to the direction-selective responses of F-mini neurons due differences in stratification. Generally, starburst amacrine cells have not been implicated in mediating the direction-selective responses of any of the highly asymmetric retinal ganglion cell types including the Fmini and JAM-B neurons (Joesch and Meister, 2016; Kim et al., 2008; Rousso et al., 2016). Second, unlike ON-OFF direction-selective neurons, the direction-selectivity of Fmini neurons is highly speed dependent, with a peak selectivity at 585 µm/s and are negligible selectivity at speeds greater than 1000 µm/s (Rousso et al., 2016). The direction-selective responses we recorded in the parabigeminal nucleus were recorded at speeds equivalent to 1500 µm/s on the retina (Figure 6). It is therefore unlikely that the Fmini neurons are contributing to this response. Finally, like Fmini, the asymmetric JAM-B neurons were originally identified as direction-selective, however, unlike ON-OFF direction-selective neurons, their direction-selectivity is not a robust property. The original authors already noted in the original paper that the direction-selectivity of JAM-B neurons is highly dependent on each neurons individual dendritic asymmetry (Kim et al., 2008), while it further experiments across different light levels revealed that the direction-selective, but not orientation-selective, responses of JAM-B cells are sensitive to light conditions (Joesch and Meister, 2016; Nath and Schwartz, 2017). We believe the many similarities between Fmini and JAM-B neurons suggest a more extensive exploration of their response properties is necessary before they are determined to be robust encoders of directional information in the visual scene. (Rousso et al., 2016).

The recent paper from Roson et al., (2019) is highly relevant for the present work and should be included in the Discussion section.

We agree. We have now included this paper in the discussion of matched response properties between retina and central brain targets (subsection “Functional responses of retinal ganglion cells and target nuclei”). Briefly the main finding of Roson et al., 2019 is that many responses in the dLGN can be explained by a linear combination of putatively innervating retinal ganglion cell responses. This is clearly not possible in our datasets. First, using the same visual stimulus as Roson et al., 2019 we see no visual responses in either the Pbg or LP. Second, others have observed this difficultly in recording responses of neurons both in the superior colliculus, and in neurons of the pulvinar that receive input from the superior colliculus, using stimuli (e.g. white noise, full field stimuli) typically used to measure functional response properties in the retina, dLGN or visual cortex (Gale and Murphy, 2014, 2016, 2018; Bennet et al., 2019; Beltramo and Scanziani, 2019). For many of the pathways passing through the superior colliculus there are highly non-linear transformations going on that do not lend themselves well to the analysis presented in Roson et al., 2019.

Reviewer #2:The authors studied the "logic" of retinal projections to the SC according to the collicular output pathways. They used transsynaptic labeling to trace the RGCs that innervate the SC neurons projecting to the LP or the Pbg, then characterized the morphology and physiology of the labeled RGCs. Their results suggest a "projection-specific" logic. They also performed in vivo recording of the LP and the Pbg to correlate the response properties of these structures with those of the RGCs that target them di-synaptically. Overall, the topic is important, and the experiments/analysis are largely appropriate. But the authors tend to over-state their findings from a limited dataset. A number of major and minor issues need to be addressed.Major concerns:1) From the RGC morphological analysis, 12 clusters were observed. According to Figure 3, ALL of them project to both structures, with different degrees of bias. But the authors over-interpret these results by stating "uniquely sampling" (Abstract). This is not supported by the data and would be extremely misleading to casual readers of the article, especially given the relatively small dataset. For example, cluster 8 in Figure 3 only contained 8 cells, and 2 of them projected to Pbg and 6 to LP, and yet it was considered one of the "clusters that are almost exclusively part of the pulvinar circuits". Similarly, the number of samples in the RGC physiology experiments are rather small for most clusters.

We agree that some of our language describing the degree of selectivity could appear overstated when comparing to one of the data figures, and that this would be misleading to a casual reader.

In the new version we have attempted to make sure the text reflects the data as accurately as possible and separated interpretations and conclusions from descriptions of the results. However, we maintain that some of the data does warrant an all-or-none description, for a few of the cell-types.

In general, four actions were taken to address the reviewer’s concerns. First, in this resubmission we have almost doubled the anatomical data included in the dataset. Second, we have added data with a third molecular marker for the four F-RGCs (Rousso et al., 2016). Third, instead of clustering the data into separate groups, we have created a decision tree that assigns our anatomical data to the EM reconstructions available in the EyeWire museum (http://museum.eyewire.org/; Bae et al., 2018). We belief this data set provides an excellent quantitative description of almost the entire retinal ganglion cell population. By augmenting this database with data from identified retinal ganglion cells in other publications (F-Cells Rousso et al., 2016; Sumbul et al., 2014; HD1/HD2/vertical OS cells Jacoby and Schwartz 2016 and Nath and Schwartz, 2017) we were able to reliably assign each dendritic tree in our dataset to one of the cell-types in this database using a limited set of hierarchical decisions based on quantifiable characteristics of the neurons, starting with the stratification patterns of the neurons. This approach has allowed us to provide a rational for the total number of cell types that exist in our dataset and positively define their types. Fourth, we have added two statistical tests to determine:

a) if a cell-type under or overrepresented in the two circuits (two-proportion z-test)

b) if a cell-type is sampled significantly by a pathway (bootstrap test)

Finally, we have adjusted our language such that statements accurately reflect the data presented. We found that each cell-type that showed a statistically negative bias to one of the circuits was also found to not innervate that circuit with enough neurons to be statistically greater than 0. If these two tests are significant, then the cell type is considered to uniquely innervate one of the circuits and not the other (Figure 2). Now the minimum number of samples in a given cluster is 11, the median is 30 and total number of neurons that we could reliably cluster is 550 (but see Figure 1—figure supplement 2). Note that doubling the size of the data set has not increased the number of cell types we found in the two circuits. In the discussion we add a paragraph about the caveats to these statements.

2) The efficacy and potential sampling bias of rabies tracing need to be addressed. This is necessary in order to conclude that the two circuits "sample from a limited set, ~14 out of more than 30".

We have added a paragraph to the Discussion section regarding the potential bias of rabies tracing. In addition, we have doubled the number of neurons included in the study and do not observe an increase in the number of cell types being sampled by these two circuits. Further, as we sample widely from both large (alpha neurons) and small neurons (F-mini, DS and HD), and all neurons are thought to form glutamatergic synapses with SC neurons we think it unlikely that a strong tropism would be the cause of the unsampled neurons. However, we do believe that the speed with which viral particles are transported back to the retina do mean our data is has an over representation of large neurons that likely does not reflect the actual ratio of innervation.

3) The use of Ntsr1-GN209-Cre mice needs to be raised as a potential technical concern. It is possible that the Ntsr1 positive cells are not the only SC cells that project to the LP, As a result, using these cre mice could under-estimate the RGC types that target SC-pulvinar pathway.

We have addressed this issue in the Discussion section. Briefly, there is evidence from both anterograde and retrograde experiments that Ntsr+ / wide-field neurons are indeed the only type innervating the pulvinar (Zhou et al., 2017; Gale and Murphy, 2016). We therefore belief that the Ntsr1-GN209-Cre mice are not biasing our results in a strong way.

Reviewer #3:Overall, the study is well motivated, well constructed, and well presented. It is really positive to see the morphology of all the RGCs in supplementary material. The manuscript primarily describes viral-mediated morphology and functional imaging of retinal ganglion cells that connect via the superior colliculus to two important brain nuclei (LP, Pbg); this morphological and functional dataset is important and impressive. In addition, the authors record extracellularly from neurons in the LP and Pbg (with unknown inputs) and measure response to several stimuli.Major points:1) The first part seems to imply that (a) there are ~12 morphological classes of RGCs projecting to the 2 pathways, a subset of the ~30 classes previously demonstrated in mouse retina, and (b) that some distinct classes of RGC project to LP or Pbg. First maybe I am missing something, but it is not clear to me if the clustering applied to this morphological data would reveal 30 classes across the retina – it may reveal 14, or it may reveal 300. Without some norm for the clustering we don't know what fraction of mouse RGC classes are identified in these analyses.

To address the lack of clarity with the clustering approach we have changed our approach. Instead of clustering the data into arbitrary groups, we have created a decision tree that assigns our anatomical data to the EM reconstructions available in the EyeWire museum (http://museum.eyewire.org/; Bae et al., 2018). We belief this data set provides an excellent quantitative description of almost the entire retinal ganglion cell population. By augmenting this database with data from identified retinal ganglion cells in other publications (F-Cells Rousso et al., 2016; Sumbul et al., 2014; HD1/HD2/vertical OS cells Jacoby and Schwartz, 2016 and Nath and Schwartz, 2017) we were able to reliably assign each dendritic tree in our dataset to one of the cell-types in this database using a limited set of hierarchical decisions based on quantifiable characteristics of the neurons, starting with the stratification patterns of the neurons. This approach has allowed us to provide a rational for the total number of cell types that exist in our dataset and positively define their types. Note that doubling the size of the data set has not increased the number of cell types we found in the two circuits.

Second there is no way to know where these RGCs project to outside the two target areas, so we do not know if for example the Pbg receives a unique sample (as implied in the Abstract) or if the Pbg pathways substantially overlap other (unsampled) pathways.

Retinorecipient neurons of the superior colliculus project to nuclei other than the Pbg and pulvinar, e.g. the dLGN, vLGN and deeper layers of the colliculus itself. In the case of LP, there is strong evidence that wide-field neurons uniquely innervate the LP and no other target (Gale and Murphy, 2014; Zhou et al., 2017). However, for the Pbg it is possible that neurons projecting to the Pbg send collaterals to other downstream targets. While this is implied in the genetic cell-type analysis performed by Gale and Murphy, (2014, 2018), preliminary data in the lab suggest that this might not be case. We have not, to date, seen co-labeling of inhibitory projection, GAD2-Cre, neurons when retrogradely labeling them from two different downstream targets the Pbg and LGN. However, in general, we do not have a good idea what proportion of collicular neurons send out axons to more than one brain region.

In the end, this question lies outside the scope of the current work. Here the aim is not to delineate all output circuits to which a given retinal ganglion cell type provided input, but to compare the inputs to two identified collicular circuits and ask whether the rules of wiring of the superior colliculus are target dependent. We know that both the colliculo-pulvinar and the colliculo-parabigeminal circuits mediate innate fear behaviors and we asked whether they would do so, at least partly, based on selective sampling of visual input from the retina.

Third, it is not clear to me from the manuscript how confident the authors are in estimating overlap in LP and Pbg RGC pathways – in Figure 3 for example, clusters 4,7 and 12 are inferred to be part of the Pbg pathway but clusters 5,9 and 11 are inferred to innervate both circuits, but these all appear to have similar biases or have so few neurons that the confidence intervals on the% to each pathway must be very large. The manuscript needs much greater clarity about how these inferences are made, and the statistical support that they have.

We have taken four actions to address this common concern among the reviewers. First, in this resubmission we have doubled the anatomical data included in the dataset (n = 599). Second, we have added data with a third molecular marker for the four F-RGCs (Rousso et al., 2016). Third, instead of clustering the data into separate groups, have created a decision tree that assigns our anatomical data to the EM reconstructions available in the EyeWire museum (http://museum.eyewire.org/; Bae et al., 2018). We belief this data set provides an excellent quantitative description of almost the entire retinal ganglion cell population. By augmenting this database with data from identified retinal ganglion cells in other publications (F-Cells Rousso et al., 2016; Sumbul et al., 2014; HD1/HD2/vertical OS cells Jacoby and Schwartz, 2016, 2016 and Nath and Schwartz, 2017) we were able to reliably assign each dendritic tree in our dataset to one of the cell-types in this database using a limited set of hierarchical decisions based on quantifiable characteristics of the neurons, starting with the stratification patterns of the neurons. This approach has allowed us to provide a rational for the total number of cell types that exist in our dataset and positively define their types. Fourth, we have added two statistical tests to determine:

a) if a cell-type under or overrepresented in the two circuits (two-proportion z-test)

b) if a cell-type is sampled significantly by a pathway (bootstrap test)

Finally, we have adjusted our language such that statements accurately reflect the data presented.

2) I was confused by how the retinal functional measurements are clustered and categorised. The authors say they identified 12 groups (subsection “Functional classes of retina ganglion cells show refined pathway selectivity”) but do not say how these were identified. For example, it is not clear to me why Group (ii) is a sustained OFF cell when the onset response appears transient, and on what basis it is distinguished from Group (iv). Some more explanation of the response to these stimuli would be useful; claims of functional differentiation between cells, and therefore whether or not particular functional classes project to particular brain areas, would also seem to require statistical support.

In the updated version of the manuscript we have decided not to cluster the physiological responses independently. Instead, we have added patched cells for which we also retrieved the cell’s morphology. These cells were assigned to the EyeWire data set, which allowed us to then show visual response properties of anatomically classified retinal ganglion cells. We used the average response properties of these anatomically classified neurons as visual response templates and followed the visual response assignment protocol similar to Roson et al., 2019 to assign all our patched neurons to an anatomical cluster. We believe this simplifies the narrative of the paper and more importantly consolidates the link between our physiological, anatomical and molecular datasets. The decision of whether a particular cell-type innervates a particular pathway is now confined to the anatomical and molecular datasets in Figure 2 and Figure 3.

3) The in vivo data though interesting is difficult to relate to the rest of the data. I do not think that it enhances this paper. e.g. (1) The authors note latency differences between Pbg and LP but there also appear to be rate differences; latency is often longer in weak responses and a fair comparison would need to match the firing rates of the two populations; (2) The authors note the presence of direction selective units in Pbg but not LP (Figure 6A) but as there are only 12 units in LP I am not sure how confident one should be in this. In addition, it is clear from other work that LP organisation depends on e.g. AP location – were the injection sites and recordings matched in location? As for the other data I could not find confidence intervals on the estimates of functional properties in Pbg and LP, making it difficult to know how well they can be distinguished.

We have addressed the reviewer’s comments in three ways. First, we have consolidated the presentation of the *in-vivo* datato focus on one difference (direction-selectivity) and one similarity (response to dark expanding discs) in the in-vivo data. We believe, this comparison demonstrates that the pathway specific wiring has the potential to explain some of the response properties of downstream targets. A discussion of the difficulty in making this link has been added to the discussion in the context of Roson et al., 2019 work showing linear combinations of retinal responses can explain some of the responses in the LGN.

We agree with the criticism of the latency comparison and have removed it. In addition, we have demonstrated that the recordings we make in the pulvinar are from the posterior portion of the pulvinar that receives input from the superior colliculus (Figure 6 and Figure 4—figure supplement 1). We have also added more evidence about the location of the injection sites (Figure 1—figure supplement 1). In the pulvinar, the precise injection site is not critical as the use of Ntsr1-GN209-Cre mouse line limits our infection to pulvinar projecting collicular neurons. The relative distribution of different functional properties in the Pbg and pulvinar are now supported with statistical tests to determine if each response feature has a biased distribution between the two circuits (two-proportion z-test).

4) The Supplementary file 1 does a good job of trying to align the classes identified by the authors with those identified by others. It is much appreciated. I think that a similar approach is necessary to be able to join the different data sets presented here – it is often unclear what the basis for deciding on the category is for a particular dimension of analysis (functional, morphological, immunostaining) and how categories are matched of categories across dimensions.

We have updated the table according to the new classification scheme that we used to analyze data and simplified the representation. We believe it is much clearer now and is presented in the main text. As we now have a single classification scheme in the paper the table is much easier to read and the basis of classification evident in the data itself.

5) There are too many unsupported and/or ambiguous phrases (non exhaustive list: Abstract: "projection specific", "uniquely sampled", "correlated well", "mechanistic basis for selective triggering of visually guided"; Introduction" "we found strong specificity"; Results section: "exclusively part of"; "very strong bias"; "rather small"; "relatively large"; "exceptionally broad"; "striking selectivity"; "had a tendency towards"; Discussion section: "clear segregation", "strong preference", "could be explained by selective sampling of different retinal ganglion cell types", "dedicated set of connections", "confidently identify", "high degree of regularity" etc). I believe the data is quite clear – and the wording should reflect the data the authors show, and the appropriate statistical analyses that they apply.

We agree that the text did not reflect the data as accurately as it should have. We have attempted to make sure the text reflects the data as accurately as possible and separated interpretations and conclusions from descriptions of the results. However, we maintain that some of the data does warrant an all-or-none description, for a few of the cell-types. We believe this is now clear in the increased dataset and clearer analysis (Figure 2). These statements are now supported with statistical tests to determine if each cell type has a biased distribution between the two circuits (two-proportion z-test), and whether each cell type is represented with significantly more than 0 cells in a given circuit (bootstrap test). We found that each cell-type that showed a statistically negative bias to one of the circuits was also found to not innervate that circuit with enough neurons to be statistically greater than 0. If these two tests are significant, then the cell type is considered to uniquely innervate one of the circuits and not the other (Figure 2). In the Discussion section we add a paragraph about the caveats to these statements.

[Editors' note: the author responses to the re-review follow.]

The reviewers have discussed the reviews with one another and the Reviewing Editor has drafted this decision to help you prepare a revised submission.One salient point emerged from the discussion among reviewers: a need to present the results about projection bias in a way more closely tied to the original data. Currently the data is subjected to a bootstrap analysis to determine if the number of projections to a given area is significantly different from 0, and the results are then summarized (e.g. in Figure 8) in an all-or-none fashion. But this does not accurately represent the fact that none of the RGC types projects exclusively to one or the other SC target area. We all agreed that the data should be presented in a more straightforward way – e.g. as numbers of projections or the ratio of those numbers (with confidence intervals). Significance tests (probably non-parametric) could be applied to whether the number of projections to the two areas differ significantly. More details about those concerns, as well as several other issues, can be found in the individual reviews below.

We thank the reviewers for their thoughtful criticisms. We have gone through the manuscript and removed as much as possible suggestions that the projection bias is an all or none phenomenon. In Figure 8 we have kept the same summary graphic but relabeled the heading to make it clear this represents a strong bias/preference rather than an exclusive sampling. We have removed our bootstrap analysis to test if the number of projections to a given area is significantly different from 0.

To present the data in a more straightforward manner, we have modified Figure 2C to represent the% difference in sampling of each cell-type by each pathway with a confidence interval and p-values estimated using a bootstrap analysis and corrected for multiple comparisons (see Figure 2C). The two statistical tests of projection bias, two-proportion z-test and bootstrap analysis, agreed with each other.

Reviewer #1:This is a revised paper about the projections from retina to SC to two SC targets: the pulvinar and the parabigeminal nucleus. The paper uses an impressive array of circuit tracing and electrophysiological approaches to show that the retinal ganglion cells that provide (via the SC) input to these two target areas differ considerably. The paper has improved considerably in revision, and the central message is very clear and well supported by the data (with one important exception – see below). The authors should be congratulated on both the work and on the strength of the revisions. I have a few suggestions below for clarity.Subsection “Some visual responses of pulvinar and parabigeminal nucleus are explained by selective innervation of retinal ganglion cell types” (and later in the Discussion section): I don't think the lack of responses to full-field stimuli requires a nonlinearity. For example, you could have a linear summation of responses with oppositely signed weights from two sets of cells with similar responses to the chirp (or other full field) stimuli. I believe that in the LGN work the ganglion cells were combined with mostly or exclusively positive signs, so I think the difference here is that you either need a nonlinearity or a combination of responses with differing signs so that you can get cancelation.

We agree and thank the reviewer for pointing out shortcoming. The text has been changed and now reads:

In the Results section: “Finally, a striking difference was observed between the responses of retinal ganglion cells innervating the different circuits and the responses of neurons in the target nuclei to full-field stimuli. The ‘chirp’ stimulus produces robust responses in most retinal ganglion cells but fails to illicit responses in either the posterior pulvinar, or the parabigeminal nucleus (Figure 6—figure supplement 1), which might be due to non-linear integration of retinal inputs or summation of opposite signed weights.”

In the Discussion section: “In addition, the colliculo-pulvinar circuit receives inputs from ganglion cells that respond well to big and fast objects (cluster 11 and shared inputs from cluster 1), but responses to such stimuli were weak or absent in the pulvinar neurons. These differences might reflect strong non-linearities in how retinal inputs are integrated, or they might be a result of balanced excitatory and inhibitory inputs that cancel each other out.”

Reviewer #2:The paper is even stronger and the authors have addressed most of my previous concerns. I think this is an important set of experiments with strong anatomical conclusions and less strong functional conclusions. The authors inferences generally reflect these, are fair and justified by the data with some small exceptions.1) In the Abstract: "These findings suggest that projection specific sampling of retinal inputs forms a basis for the selective triggering of behaviours by the superior colliculus". I don't think that this statement is sufficiently supported by the work, as the functional distinction is not clear – for example, the authors show that despite different retinal inputs, looming stimuli apparently activate both Pbg and LP pathways (not obviously consistent with the selective triggering of behaviours), while the potent chirp stimuli for the retinal ganglion cells apparently have no counterpart centrally and probably not behaviourally. I would think this needs rewording to offer speculation not conclusion (e.g. "These findings open the possibility that projection specific sampling of retinal inputs helps form a basis for the selective triggering of behaviours by the superior colliculus"). Similarly, the first paragraph of Discussion section accentuates the difference populations that are sampled, and ignores the similarities, creating the unfortunate impression that they are non-overlapping inputs.

We thank the review for this clarifying suggestion and have changed these sections accordingly. The text now reads:

In the Abstract: “These findings open the possibility that projection specific sampling of retinal inputs forms a basis for the selective triggering of behaviors by the superior colliculus.”

In the Discussion section: “First, the colliculo-parabigeminal and colliculo-pulvinar circuit together sample from a limited set (14 of 37) of retinal ganglion cell types (Bae et al., 2018). Second, there is a clear preference in the set of retinal ganglion cell types providing input to each circuit. While 4 putative ganglion cell types show a strong preference for the colliculo-parabigeminal circuit, and 4 others for the colliculo-pulvinar circuit, 6 other types are more equally sampled by both circuits. Third, some response properties of neurons in downstream targets can be explained by the different and shared sampling biases of each retinal ganglion cell type by each collicular output pathway, respectively.”

2) Subsection “Functional properties of retinal ganglion cells support anatomical classification” is less easily read than the other sections. I think that the authors have reconstructed 23 dendritic fields, assigned each to one of the classes, averaged the responses within each class, then assigned the non-reconstructed neurons to the same class on the basis of the similarity in their functional properties. There is some risk of circularity here, depending on the question. I am not too worried about that here, but I do think the authors need to be careful in presenting the physiological data because it may generate more certainty in anatomical-physiological correlations than is warranted. This can be circumvented by making it clear in the legends to Figure 4 and Figure 7. Also, the 'N=' values on the left of the rows in Figure 4 should be adjusted to report the number of anatomically identified units in each cluster to make sure there is no confusion – the total N could be reported in the physiology column. N values should also be reported in Figure 7.

We agree with the reviewer’s criticism, though the number of reconstructed patched neurons is 48 not 23. This has been corrected in the text and figure. In addition, we have made two fundamental changes to clarify the figures and text. First, in Figure 4 and Figure 7 two values are presented. n_a+p_ and n_total_ to represent to number of cells with both anatomy and total number included, respectively. In addition, we have removed some of the language reporting consistency of responses within a cluster.

One point- I can't quite work out the numbers here – in subsection “Assigning recorded retinal ganglion cells” the authors state that 11 clusters had at least 1 cell with anatomy and physiology, then state they measured chirp responses of 123 patched cells (should this be 23?), and then state that they further analyse only clusters with at least 4 patched cells. In Figure 4 there 7 clusters (minimum number of units == 28). I therefore assume that the authors mean that the anatomical-physiological correlation of a cluster can be based on <4 patched cells with both anatomical and physiological measurements. If not, please clarify, if so, please make that clear, and simply state the numbers of cells with combined a/p in each cluster.

We have added further explanations to both the main text (Subsection “Retinal inputs to the parabigeminal and the pulvinar circuit differ in molecular signature” and subsection “Assigning recorded retinal ganglion cells”). Briefly, we have recorded visual responses from 123 cells. Out of these 123 cells, we retrieved the anatomy of 48 cells and classified those cells during the morphological classification step (Figure 2). Based on this classification, we had patched at least 1 cell in 11 clusters. For each anatomical cluster with a patched neuron we used the responses of its assigned neurons to create a response template that was then used to assign the remaining 75 patched cells without anatomical data to one of the 11 clusters with combined anatomy and physiology data. Each cluster (of the 11) that had at least 4 assigned cells was used in Figure 4 and Figure 7, which contains 7 clusters. Taken together, this means we report on 93 of our 123 patched cells assigned to our anatomical clusters in Figure 4 and Figure 7.

3) Subsection “Retrograde transsynaptic labelling of retinal ganglion cells”: Some moderation on the Ntsr1 line is still needed. The Zhou et al., (2017) study cited shows fairly clearly only WF-like cells, probably though not certainly Ntsr1, in their Figure 2B but that is a contralateral section (ie. showing the contralateral projection to LP). Additional more superficial cells, which may or may not be WF, appear to be labelled in e.g. Figure 8C after LP injections. I've not strong opinions about whether there are or are not multiple subclasses of SC neurons (including multiple subclasses of WF neurons) but I would argue that we don't yet know with certainty.

We have added the observation based on Figure 8C from Zhou et al., 2017 to the text (subsection “Retrograde transsynaptic labelling of retinal ganglion cells”): “In addition, unbiased retrograde labelling of collicular neurons, using HSV, from the pulvinar has predominantly revealed wide-field neurons, though a small number of neurons that might be of a different type were also seen (Zhou et al., 2017).”

Reviewer #3:The authors have significantly revised the manuscript by adding more data, performing new analysis, and including necessary discussion. The manuscript is much improved as a result. My only remaining issue is the "all-or-none" statement regarding projection specificity. I disagree with the new bootstrap analysis used to support their conclusion and I am skeptical of the procedure to analyze percentage instead of the absolute numbers. My concerns are elaborated below.Using "cluster 3" as an example, 11 cells were found, with 8 in Pbg experiments and 3 in LP experiments, out of the 196 to Pbg and 354 to LP (by the way, was there 550 or 599 total cells in the data set? The numbers in the manuscript are inconsistent).From these numbers, the authors calculated the percentage to be ~4% to Pbg and <1% to LP. I agree that there appears to be a preference for RGCs in this cluster to project (indirectly via SC) to Pbg, and I also think such a result is worthy of a publication. But the authors went with a bootstrap analysis that shows the percentage to the LP is NOT different from 0, thus stating they "almost exclusively" project to Pbg (and complete exclusivity in the summary figure in Figure 8). This is just wrong -- cells in this cluster clearly project to LP. This statistical test is just inappropriate, and Figure 8 is dangerously misleading. Why not just illustrate the preference with lines of different thickness?Regarding calculating the percentage, I am not sure that's appropriate or informative either. The percentage is out of the total number of labeled cells, not the likelihood of a certain type RGCs projecting to one structure over the other. Again using cluster 3 as an example, the 8 vs. 3 difference (# of labeled cells) indicate that these cells are more than twice likely project to Pbg, from the RGCs' perspective in terms of "projection logic", but not quite the 4 times difference as indicated by the percentages. Additionally, the authors acknowledge that the labeled cells in LP experiments were consistently more than in the Pbg experiments, and that this could be due to the fact that Pbg was difficult to target (Subsection “Retrograde transsynaptic 364 labelling of retinal ganglion cells”). In other words, the percentage difference could likely be affected by experimental factors.

We thank the reviewer for their thoughtful criticism.

First, we labeled 599 neurons (Figure 1), of which we were able to classify 550 (Figure 2). We have added some clarification in subsection” Biased sampling of retinal ganglion cell types by the colliculo-parabigeminal and the colliculo-pulvinar circuit”:

“This analysis revealed that 14 of the 37 classes of retinal ganglion cells contained at least 1% of the ganglion cells from our data set, suggesting that a limited set of retinal ganglion cell types are sampled by the colliculo-pulvinar and colliculo-parabigeminal circuits (Figure 2 and Figure 2—figure supplement 1). These 14 putative cell types contain 550 out of the 599 classified cells and will subsequently be referred to as cluster 1-14…”

Second, regarding the all-or-none statements. We have replaced all all-or-none like statements with statements about preference or bias in the Abstract, Introduction and Results section.

Third, we fully agree with the criticism of the bootstrap analysis used to support the all-ornone analysis. This has been removed. A bootstrap analysis is now used to provide estimates of confidence intervals in Figure 2C and to provide a second estimate of p-values for measurements of preference.

Fourth, in Figure 8, while we have left the illustration itself the same, we have changed the labels to indicate that the lines represent strong preferences not exclusive projections. We found that adding more lines made the illustration unwieldly and very difficult to parse.

Finally, we would like to stick with the proportional representation per circuit as this more accurately represents the expectation of finding these cell-types, if similar experiments are performed in the future. We believe basing the analysis on the total number of cells counted would mean the result depend on how many total cells we counted per circuit (Imagine if we had imaged 10000 in one pathway and 1000 in the other). Our data set is quite sparse where the mean number of cells traced per retina was 6.8 for the LP circuit and 4.9 for the Pbg across 92 retinas. This difference is not the consequence of how many cells are labeled during any one experiment, but how many were isolated from their neighbors and thus easy to trace. Most of the labeled neurons in each retina were not reconstructed as the labeling was too dense.

We believe comparing the molecularly labeled cell counts in Figure 3 and Figure 1—figure supplement 3 to the classified anatomical types in Figure 2 illustrates that the proportion of labeled neurons is a better representation than the total number. For the putative DS cells (cluster 4 and CART labeled) the relative proportion of labeled neurons in our anatomical dataset is 8% in Pbg and 0.2% in LP, equivalent to a ~40 times more likely projection to the Pbg circuit. This matches well with the 7% and <.01% in our molecular count. However, if we had taken the total numbers the anatomical data suggest only a 16 times increase in expectation. A second comparison is the proportion of sON-alpha neurons found counting anatomical types or molecular types. In Figure 3 we found it to be more than 20 times more likely to find sON-alpha cells in the Pbg circuit. This is a better match to our comparison of proportions shown in Figure 2 (~10x more likely in Pbg circuit) than a comparison of total numbers would be (4.5x more likely in Pbg circuit).

However, although we believe the proportional representation better reflects the biology, our analysis of preference presented in Figure 2C is not qualitatively affected by using the total number of counted cells. Author response image 1 is a version of Figure 2 with an additional column that repeats the analysis done in Figure 2C but with total counts.
